



# A zooplankton diel vertical migration parameterization for coastal marine ecosystem modeling

Ariadna Celina Nocera[1,2], Dany Dumont[3], and Irene R. Schloss[4,5,6]

[1]Centro para el Estudio de Sistemas Marinos (CESIMAR), CCT CENPAT-CONICET, Puerto Madryn, U9120ACD, Argentina.
[2]Universidad Nacional de la Patagonia San Juan Bosco, Puerto Madryn, U9120ACD, Argentina.
[3]Institut des sciences de la mer de Rimouski (ISMER), Université du Québec à Rimouski (UQAR), Rimouski, Québec, G5L 3A1 Canada.
[4]Instituto Antártico Argentino, Buenos Aires, Argentina.
[5]Centro Austral de Investigaciones Científicas (CADIC), CONICET, Argentina.
[6]Universidad Nacional de Tierra del Fuego, Ushuaia, Argentina.

**Correspondence:** Ariadna Celina Nocera (anocera@cenpat-conicet.gob.ar)

**Abstract.** A simple parameterization of zooplankton vertical swimming is proposed as a way to reproduce the diel vertical migration (DVM) behavior, which refers to the daily descent of aquatic organisms hundreds of meters below the surface at dawn and their return to the surface at dusk, a phenomenon that is widespread among most zooplankton species. The swimming behavior is mechanistically parameterized as a function of the local irradiance and food availability, and is incorporated in a

simple biogeochemical model coupled with a water column turbulence model in an Eulerian framework. The DVM behavior and its impact on plankton dynamics are investigated in an idealised configuration representing a marine coastal ecosystem. The sensitivity of the model to key parameters such as the zooplankton swimming speed, grazing rate, the optimal irradiance and turbulent diffusivity is evaluated with respect to three metrics representing the actual DVM behavior, the zooplankton-to-phytoplankton grazing coupling efficiency, and the vertical carbon export. Results show that the parameterization is able to

reproduce the main characteristics of present knowledge about zooplankton DVM, and that the associated ecosystem responses are strongly sensitive to the maximum grazing rate, and moderately sensitive to other parameters.

## 1    Introduction

Marine ecosystems are crucially important for determining Earth's climate (Le Quéré et al., 2005) as they affect ocean bio-

geochemistry and carbon cycling (Heinle and Slawig, 2013), but also because they remove part of the anthropogenic $CO_2$ from the atmosphere (Le Quéré et al., 2018; Sarmiento and Gruber, 2002). Carbon dioxide is first diluted in sea water and then stored into organic matter through photosynthesis by phytoplankton in surface waters. This particulate organic carbon (POC) can then be grazed by zooplankton and/or sink to depth where it accumulates on the seabed, is consumed by benthic





macro- and microorganisms and finally buried in sediments. (Boyd and Stevens, 2002; Longhurst, 1991; Turner, 2002). Many biological, physical and biogeochemical processes thus act to modulate how carbon is exported from the surface to the bottom of the ocean. Coastal seas and shelves are areas of great importance for carbon sequestration and other ecosystem services due to their high biological productivity (Holt et al., 2009; Simpson and Sharples, 2012).

The efficiency of the biological carbon pump is regulated by zooplankton and micronekton vertical migration (Bianchi et al., 2013; Steinberg et al., 2000, 2002). Zooplankton, other than transferring energy to higher trophic levels, plays a central role in the active transport of dissolved and particulate organic matter to depth (Ducklow et al., 2001). The production of feces is particularly efficient for exporting carbon to the ocean's floor as fecal pellets sink relatively fast, in the order of $1\text{-}10^2$ meters per day. The rate and timing at which they are produced is thus potentially an important factor modulating the efficiency of the

carbon pump (Turner, 2015).

Diel vertical migration (DVM) refers to the daily descent of marine organisms hundreds of meters below the surface at dawn and their return to the surface at dusk, a phenomenon that is widespread among most zooplankton species (Ringelberg, 2010). It is also a mechanism by which the organic matter is removed from the ocean surface layer towards the water column interior (Longhurst and Harrison, 1989; Longhurst, 1991). Although causes of this collective behavior are not completely understood,

evading visual predators and searching for food are thought to be the most important drivers of DVM (Pearre, 2003).

Even if the effect of zooplankton DVM on carbon fluxes depend on the biomass and the species of the migrating organisms, several processes have been shown to provide large quantities of nutrients below the euphotic zone and to contribute to carbon export. Among them, excretion and respiration (e.g. Steinberg et al., 2000), fecal pellets production (Laurenceau-Cornec et al., 2015), death and molt (Morales, 1999; Zhang and Dam, 1997) as well as the lipidic pump (Jónasdóttir et al., 2015; Record

et al., 2018; Visser et al., 2017) were identified.

Given the key role zooplankton plays in the marine environment, it is important to consider and evaluate the inclusion of the DVM behavior in models. To date, only few modeling efforts have included and studied DVM representations in biogeochemical models. For example, Maps et al. (2011) proposed a detailed model based on the life history of copepod *C. finmarchicus*, including ontogenetic and diel vertical migrations as well as their interactions with hydrological and atmospheric

forcings. Their DVM behavior was prescribed as a function of the clock time over a 24 h cycle and was thus constant throughout the year. Although heterogeneous distributions were obtained for one copepod specie, interactions with other ecosystem components were left out of their analysis. Bianchi et al. (2013) implemented a generic model for zooplankton (including micronekton) based on metabolic features, and focused on the decoupling of processes involved in the nutrient cycles (feeding, respiration and excretion).

DVM is intimately related to ocean's physics. In calm conditions, i.e. when turbulent diffusivities are low, zooplankton may migrate at a rate determined by their swimming speed, while in more energetic conditions, they may not be able to fight against turbulent eddies and despite their efforts to migrate, be diluted due to intense vertical mixing. At the same time, turbulent diffusivities and ocean's stratification affect nutrient and phytoplankton distributions, which in turn affect primary and secondary production. Therefore, the effects of DVM on secondary production and on vertical carbon fluxes are non-

linearly modulated by coupled biogeochemical and physical processes linked to ocean's turbulence and stratification. To our





knowledge, the effect of turbulence and stratification on plankton dynamics with DVM has not been studied. It is our goal in this paper to characterize if, in which conditions, and how the migratory behaviour of zooplankton impact i) the diel vertical migration as a collective and observable phenomenon, ii) secondary production, and iii) vertical carbon fluxes.

To do this we implement a simple yet realistic DVM parameterization in an Eulerian nitrate-phytoplankton-zooplankton-detritus (NPZD) biogeochemical model coupled to a water column turbulence model. We then use this model to characterize the impact of DVM on the ecosystem response, focusing on the sensitivity to turbulent mixing and grazing efficiency, as well as on the vertical export of carbon. We consider that such a characterization is necessary before including this behavior into more complex models or applied to more realistic configurations as it may affect other parameterizations, but also to facilitate model results interpretation.

The paper is organized as follows. Section 2 describes the DVM parameterization (2.1), the biogeochemical model (2.2) and both the idealized and realistic numerical experiments carried out to characterize the ecosystem response (2.3). Metrics used to characterize the ecosystem responses are described in section 2.4. Results are presented in section 3 and discussed in section 4. A conclusion is presented in section 5.

## 2 Model description

### 2.1 Diel vertical migration (DVM) parameterization

Vertical migration has been studied considering zooplankton ontogeny or physiological factors (Aita et al., 2003; Batchelder et al., 2002; Maps et al., 2011). Elucidating the causes of this migration or establishing significant correlations with environmental or physiological conditions was at the heart of these papers' discussions.

The parameterization we propose here is not meant to be a faithful representation of how Nature works. It is rather meant to model zooplankton swimming behavior as observed in a wide range of marine and freshwater systems and that is based upon mechanistic rules that are simple enough so that their effects can be relatively easily interpreted. Zooplankton vertical swimming behavior we impose here depends on light intensity in the water column and food (phytoplankton) availability. It is generally assumed that zooplankton migrate mainly to find refuge in the dark where predators are less abundant and cannot easily locate them through vision (Pearre, 2003; Ringelberg, 2010). We can model this behavior assuming that zooplankton swims towards an optimal light intensity at a speed that is maximal when the difference between the local irradiance and the optimal irradiance is large, and decreases as this difference decreases. We also assume that zooplankton will swim actively towards the preferred irraidance unless there is food in sufficient quantity. In such a case, it stops and grazes. Mathematically, the zooplankton vertical swimming velocity $w_Z$ can be expressed as

$$w_Z(z) = \begin{cases} -w_Z^{\max} \tanh\left[\log\left(\frac{I(z)+\epsilon}{I_c}\right)\right] & \text{if} \quad P(z) < P_{\min} \\ 0 & \text{otherwise} \end{cases}. \tag{1}$$

where $w_Z^{\max}$ is the absolute maximum swimming speed, which depends on the organism size, $I(z)$ is the irradiance intensity at depth $z$, $\epsilon \ll I_c$ is a small positive irradiance value that guarantees that the logarithm is defined when $I(z) = 0$, $I_c$ is the optimal

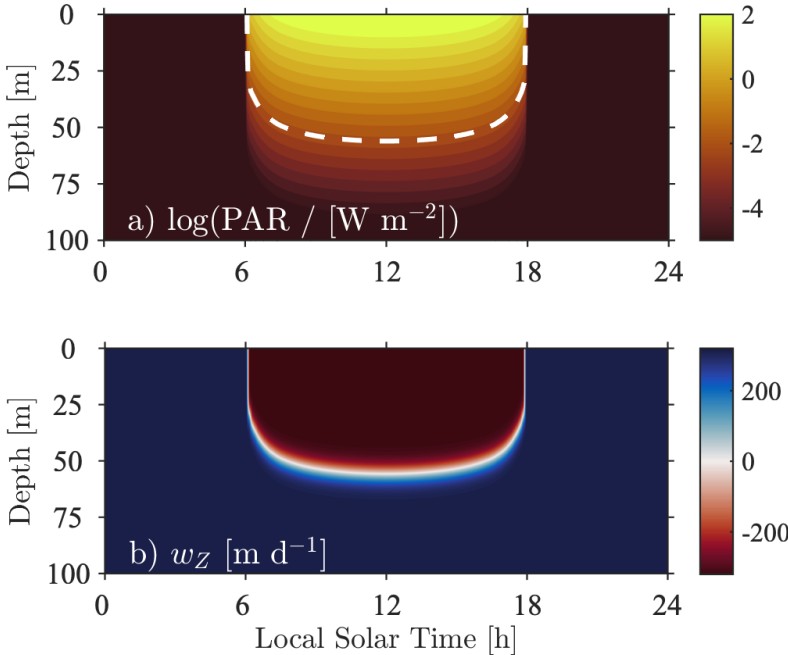

**Figure 1.** a) Example of a subsurface photosynthetically available radiation (PAR) field (in logarithmic scale) as a function of depth and local solar time at the equinox at $45°$ latitude. The optical depth is $z_d = 0.5$ m and the optimal isolume is $I_c = 0.1$ W m$^{-2}$. b) Zooplankton vertical migration velocity given by Eq. 1 associated with the PAR field in panel a), with a maximum migration speed of 320 m d$^{-1}$.

irradiance, and $P_{min}$ is the minimum phytoplankton concentration zooplankton needs to stop from migrating. Figure 1 shows an example of how $w_Z$ varies with depth and time according to Eq. 1 with $I_c = 0.1$ W m$^{-2}$, and $w_Z^{max} = 320$ m d$^{-1}$ during a diurnal cycle with maximum surface PAR at noon. The vertical swimming speed of zooplankton asymptotically reaches a maximum value when the local irradiance $I(z)$ is far from an optimal isolume $I_c$. It is positive (upward) when $I(z) < I_c$ and 5 vice-versa. An exception to this rule is that zooplankton will not swim if the local food (phytoplankton) abundance is high, i.e. larger than some threshold concentration value $P_{min}$.

## 2.2 Biogeochemical model

The DVM parameterization is applied to the zooplankton compartment of a NPZD-type marine ecosystem model that is coupled to a water column turbulence model, following Burchard et al. (2006). The biogeochemical model is based on Kühn and Radach 10 (1997) and represents a nitrogen-based planktonic system that includes nitrogen recycling through a microbial loop. Figure 3 shows the model variables as well as the conceptual illustration of the DVM and its effects on the active transport of carbon and on the coupling intensity between primary producers and grazers.

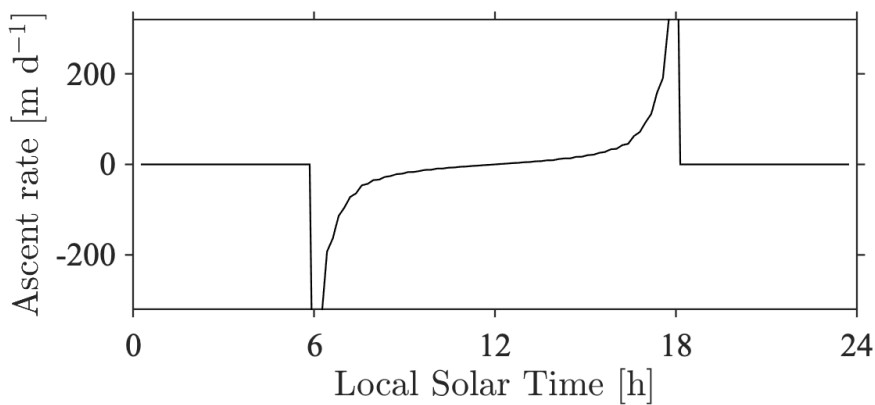

**Figure 2.** Ascent rate (positive upward) of the isolume $I_c = 0.1$ W m$^{-2}$ in the water column as a function of time for the light profile of Fig. 1.

Nitrate (NO$_3^-$) is the main limiting nutrient considered in the model. Primary production is provided by one phytoplankton group (PHY), which represents phytoplankton in the 10 to 200 $\mu$m size range, corresponding to the optimal food size for mesozooplankton, particularly for copepods (as a group) (Fuchs and Franks, 2010; Sabatini et al., 2012). Phytodetritus, dead zooplankton, remains of grazing and fecal pellets are undifferentially included in a detrital compartment (DET). The excretion

and respiration of organisms can replenish the biogeochemical model with nitrogen through the microbial loop. This recycling process includes three state variables, namely bacteria (BAC), ammonium (NH$_4^+$) and labile dissolved organic nitrogen (DON). The seven state variables $c_i$ are expressed in the model as nitrogen concentrations (mmolN m$^{-3}$), which are later converted for analysis to carbon concentrations (mgC m$^{-3}$), assuming a constant carbon-nitrogen ratio of 6.6:1 for phytoplankton and detritus, and 7.0:1 for zooplankton (Redfield, 1958). They obey advection-diffusion-reaction equations of the form

$$\frac{\partial c_i}{\partial t} = -\frac{\partial (w_i c_i)}{\partial z} + \frac{\partial}{\partial z}\left(K_z \frac{\partial c_i}{\partial z}\right) + \text{reactions} \tag{2}$$

where $K_z$ is the vertical turbulent diffusivity, and $w_i$ are the vertical advection velocities of state variables $i$ with respect to water. Note that the advection term is here expressed as the gradient of an advective flux in the case where the vertical velocity varies with depth. Phytoplankton and detritus sink at constant velocities $w_P$ and $w_D$ (see Table 1), while $w_Z$ varies with time and depth and is given by Eq. 1. Reactions refer to biogeochemical interactions between compartments. They are described

in detail in Burchard et al. (2006) (their appendix C). Here we only recall that the zooplankton grazing function follows a




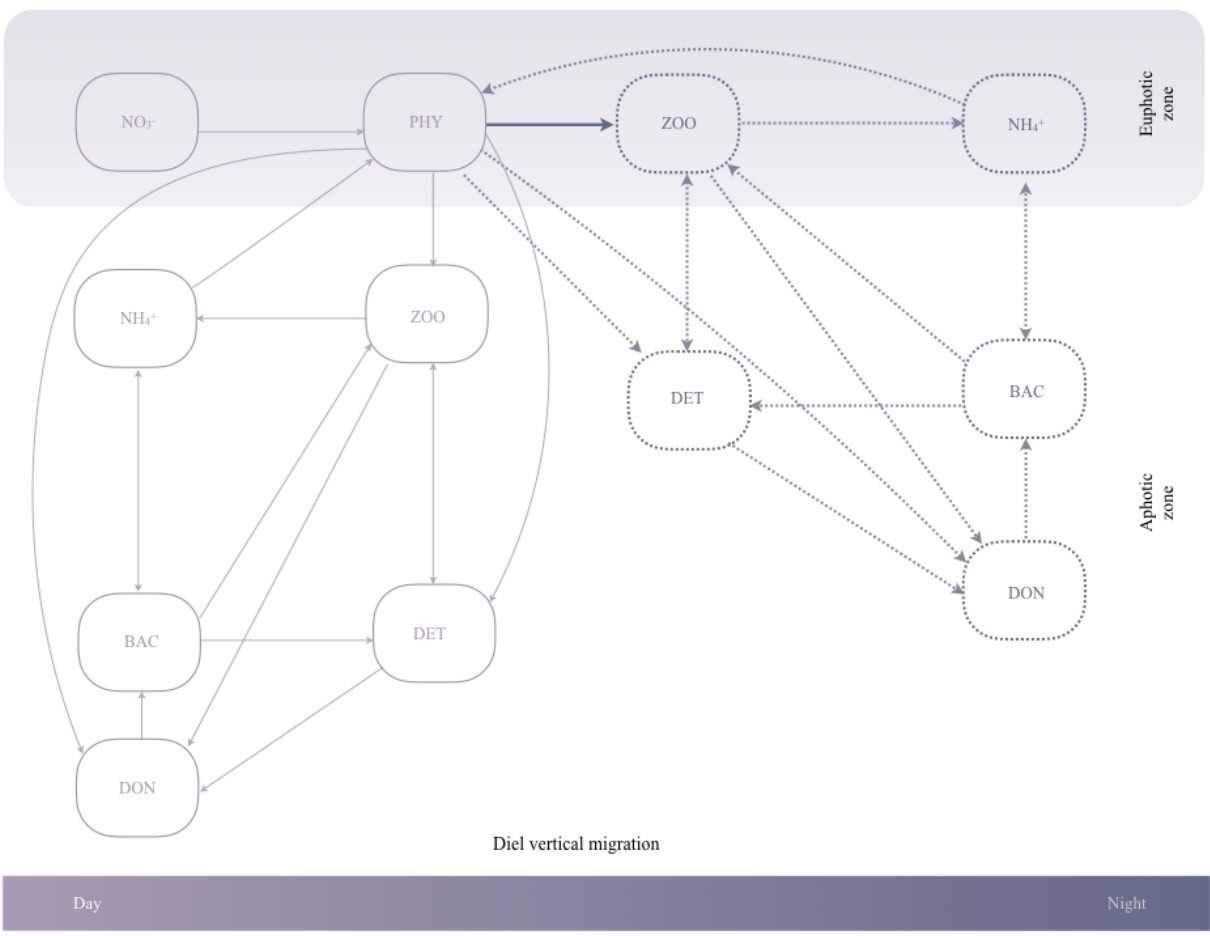

**Figure 3.** Conceptual model with seven state variables (see text) adapted from Burchard et al. (2006) with the inclusion of diel vertical migration for the zooplankton group. Modeled food web and interactions are shown during the day (filled lines) and at dusk (dotted lines). The thick arrow represents a strong relation between compartments.





sigmoidal form where $g_{\max}$ is the maximum grazing rate, that we will vary, and $k_g = 1.0$ mmolN m$^{-3}$ is the half-saturation constant that remains unchanged for all the experiments (section 2.3).

Zooplankton (ZOO) secondary production and DVM are parameterized for copepods, which is the most common zooplankton group in the ocean (Turner, 2004). No specific species are explicitly represented. Marine copepods are well known as diel

vertical migrators in several areas (e.g. Longhurst, 1991). They spend the nighttime in surface waters looking for food and retreat at depth during daylight hours as they become more vulnerable to visual predators (Fig. 3). Their migration is determined by their maximum swimming speed, which depends on their body size range that lies between 0.2 and 20 mm.

### 2.3    Numerical experiments

The model configuration we use is a 100-meter deep water discretized in 50 vertical levels with thicknesses that vary from 1.1 m

near the surface and bottom boundaries to 2.6 m at mid-depth. The ocean is initialised with a two-layer stratification profile for nutrients (see Fig. 5) that is restored with a two-week relaxation time. This allows to set a mixed layer depth that remains similar on average when surface heat and momentum fluxes are modified from one simulation to another. Nitrate concentrations are set to 0.3 mmolN m$^{-3}$ in the mixed layer and 15 mmolN m$^{-3}$ in the deep layer, and restored with a 6-month timescale. Solar radiation that includes the diurnal cycle typical of the mid-latitude (45°) is prescribed for all simulations and provides

the photosynthetically available radiation (PAR) in the water column. The turbulent diffusivity profile is calculated using the $K$-profile parameterization (KPP) of Large et al. (1994) in response to a prescribed surface wind stress. Prescribed values are listed in Table 2 and examples of associated turbulent diffusivity profiles are shown in Fig. 5.

To characterize the impact of DVM on ecosystem dynamics, we proceed by carrying a systematic sensitivity analysis of a number of ecosystem indicators with respect to model parameters we expect may interfere with the migratory behavior.

Some parameters are directly involved in the DVM parameterization, namely the maximum migratory speed $w_Z^{\max}$, the critical phytoplankton concentration that stops the migration $P_{\min}$ and the critical light level $I_c$ at which zooplankton settles. Other parameters may also affect the migratory behavior or its overall effect on the ecosystem. The maximum grazing rate of zooplankton on phytoplankton is one of them. The DVM parameterization may decouple zooplankton from its food source, thus decreasing the overall grazing efficiency, or strengthen the coupling if zooplankton is able to swim towards phytoplankton.

Grazing efficiency, controlled by the maximum grazing rate $g_{\max}$, may thus be significantly altered by the DVM. Similarly, turbulent mixing may influence the migratory behavior as well as the vertical patchiness of biogeochemical tracers in the water column.

### 2.4    Metrics

Despite the relative simplicity of the biogeochemical model, the proposed DVM parameterization and the model configuration,

the parameter's space is still very large. In order to characterize the actual zooplankton migratory behavior and its impact on the ecosystem, we define some metrics, or indicators, that are representative of key aspects of the ecosystem and we look at how these metrics are distributed in the parameter's space defined in the previous section.





The first indicator is meant to determine whether the active swimming behavior of the zooplankton prescribed by the model, triggered by non-optimal irradiance, leads to actual vertical migration. In an ideal case where turbulent diffusivity is low and phytoplankton concentrations are low (below $P_{\min}$), zooplankton should aggregate at the depth of the optimal irradiance and should follow it closely as it ascends and descends the water column over a diurnal cycle. By contrast, in turbulent conditions,

zooplankton my be diluted in the mixing layer despite its effort to migrate. Alternatively, it could remain at a depth where there is plenty of food despite non-optimal irradiance. The DVM indicator that we build here includes the two important aspects of the DVM, namely the accumulation of zooplankton around the optimal light level, and the vertical excursion of this accumulation over a diurnal cycle. First, the position $z_p$ and width $\sigma_p$ of local zooplankton maxima are obtained with the Matlab function `findpeaks` at each time step, when the vertically integrated zooplankton biomass is >1 mmolN m$^{-2}$. The

DVM metric $\Omega$ is then defined as the ratio between the vertical distance between the daily extrema of the zooplankton patch position $\Delta z_p$, and the vertical spread (or width) of the patch $\sigma_p$. When more than one patch is found, we select the patch with the largest prominence. Then we compute the daily DVM metric as

$$\Omega = \frac{\Delta z_p}{\sigma_p}. \tag{3}$$

The second metric, or indicator, quantifies the spatial coupling between zooplankton and phytoplankton. It is defined as the

proportion of the vertically integrated zooplankton biomass that is associated with phytoplankton concentrations larger than $P_{\min}$ (see Eq. 1), and is expressed as

$$\Psi = \int\limits_{-H}^{0} \mathcal{M} Z(z)\, dz \left[ \int\limits_{-H}^{0} Z(z)\, dz \right]^{-1} \tag{4}$$

where $\mathcal{M}$ is a mask function equal to 1 if $P(z) \geqslant P_{\min}$ and 0 otherwise. If zooplankton migrates efficiently, the entire population should be concentrated around the optimal light level unless there is so much food that it stops migrating and eats.

The third metric represents the carbon export ($\Phi$), and is defined as the proportion of the vertically-integrated organic carbon that is found below 80 m. It is defined as

$$\Phi = r_{\mathrm{C:N}} \int\limits_{-H}^{80\ \mathrm{m}} N_{\mathrm{tot}}(z)\, dz \left[ \int\limits_{-H}^{0} N_{\mathrm{tot}}(z)\, dz \right]^{-1}. \tag{5}$$

where $N_{\mathrm{tot}}$ is the sum of the seven model variables and $r_{\mathrm{C:N}} = 7$ mmolC mmolN$^{-1}$ is the Redfield carbon to nitrogen ratio. When detritus reach the ocean floor, they are not removed or sequestred. They rather remain in the last grid cell, available for

degradation in dissolved organic carbon and further transformed in ammonium by bacteria. Detritus can also be grazed by zooplankton, which happens in the control model where zooplankton does not migrate. There, a non-realistic *benthic* zooplankton community emerges and grows as detritus accumulates (see Fig. 4a). As here we are interested in the total organic carbon export to the ocean floor, and not specifically in long term sequestration in sediments, we sum over all model compartments in our model.





The three metrics, or indicators, defined in this section are computed at each model time output. In order to end up with one value for each model run, we take the indicator time series temporal mean. Results are mapped on sections of the parameters' space explored in the sensitivity analysis.

We examine $\Psi$, $\Phi$ and $\Omega$ as a function of zooplankton grazing rate ($g_{\mathrm{max}}$), zooplankton maximum swimming speed ($w_Z^{\mathrm{max}}$),

as a function of critical irradiance ($I_c$) and a range of turbulence values (Figure 5).

## 3 Results

### 3.1 Examples of idealized experiments

To illustrate zooplankton dynamics with the diel vertical migration parameterization, we first show how zooplankton evolves over an arbitrarily chosen two-day period in multiple cases. Figure 4$a$ shows zooplankton distribution in a typical NPZD model

without DVM, i.e. associated with its food. Here, zooplankton is located in the near-surface layer, where phytoplankton grows the most, at intermediate depths associated with detritus and bacteria (also zooplankton food), and near the bottom where detritus and bacteria accumulate. This *benthic* layer is often taken off in biogeochemical models by sequestring detritus as they reach the ocean floor, preventing zooplankton from growing. By adding and switch on an active swimming behavior for zooplankton, this artifact is removed without any other modifications. Panels *b-f* show how zooplankton evolves during the

same two-day period when DVM is activated. Here we chose $I_c = 0.01$ W m$^{-2}$, $P_{\mathrm{min}} = 1.4$ mmolN m$^{-3}$ and $w_Z^{\mathrm{max}} = 320$ m d$^{-1}$, with values of $g_{\mathrm{max}}$ and wind conditions indicated on each panel. Even though zooplankton is able to swim as fast as 320 m d$^{-1}$, it does not migrate in the case shown in Fig. 4$d$ because it is always matched with phytoplankton concentrations greater than $P_{\mathrm{min}}$ (here set to 1.4 mmolN m$^{-3}$). The grazing rate is not sufficiently large to deplete phytoplankton, which remains abundant enough to provide zooplankton for a *reason* (with respect to the parameterization) to remain at this depth.

Fig. 4$b$, $c$, $e$ and $f$ show examples where DVM clearly happens: zooplankton follows the isolume $I_c$, indicated by the white line. When zooplankton approaches the surface, its distribution is affected by the presence of food and by turbulence. For intermediate grazing rate values, zooplankton shortly stops migrating around 20 m (Fig. 4$b$ and $e$), while it reaches the surface when zooplankton is able to graze and deplete phytoplankton below the critical $P_{\mathrm{min}}$ value.

Turbulence, here induced by wind shear at the ocean's surface, can act to dilute zooplankton over the mixing layer, which in

the shown example is 15 to 20 m deep. When turbulent diffusivity is low enough, zooplankton aggregate into narrow patches (Fig. 4$c$ and $f$), while if it is strong enough, it will be mixed homogeneously (Fig. 4$b$ and $e$).

The examples described show how the mechanistic parameterization of the light-driven diel vertical migration behave in various cases. Those examples are identified by red stars in the figures presented in the following section, where the parameters'space is explored with respect to the three indicators defined in section 2.4 for assessing model sensitivity.





**Table 1.** List with descriptions and values for the parameters used in the model.

| Symbol | Description | Value | Unit |
|---|---|---|---|
| $\Delta z$ | Resolution of the grid | $[1.1 - 2.6]$ | m |
| $\Delta t$ | Time step | 240 | s |
| $P_{\min}$ | Minimum phytoplankton concentration | $[0.35, 0.7, 1.4]$ | mmolN m$^{-3}$ |
| $g_{\max}$ | Zooplankton maximum grazing rate | $[0.5, 1.0, 2.0, 3.0, 4.0]$ | d$^{-1}$ |
| $w_z^{\max}$ | Zooplankton maximum swimming speed | $[0.0, 10.0, 50.0, 100.0, 200.0, 320.0]$ | m d$^{-1}$ |
| $I_c$ | Critical irradiance | $[0.01, 0.1, 10.0]$ | W m$^{-2}$ |
| $H$ | Depth of the water column | 100 | m |
| $k_d$ | Light attenuation constant | 0.3 | m$^{-1}$ |

**Table 2.** Description of numerical experiments.

| | Turbulent regime |
|---|---|
| 1 | No wind stress |
| 2 | Constant wind stress $\tau_a = 0.1$ N m$^{-1}$ |
| 3 | Constant wind stress $\tau_a = 0.3$ N m$^{-1}$ |
| 4 | Realistic wind stress without DVM |
| 5 | Realistic wind stress with DVM |

## 3.2 Sensitivity analysis

Here we describe how the three metrics vary as a function of physical and biological parameters. The DVM metric $\Omega$ indicates if zooplankton aggregates in patches around their preferred light level and if these patches truly migrate vertically during a 24 h cycle. Even though the parameterization forces zooplankton to swim towards a critical light level, food distribution and

turbulent diffusion can perturb that behavior and disrupt the migration, as shown in the previous section.

We selected parameter values for the model input based on the literature. $P_{\min}$ parameter takes values from 0.35 to 1.4 mmolN m$^{-3}$ considering average phytoplankton concentrations for conditions before, during and after the late phase of a bloom in coastal marine ecosystems (Schalles, 2006). Zooplankton grazing rate is a parameter which has been studied for decades, and we chose different rates of $g_{\max}$ in relation to the different capacities that organisms have to consume their prey. Important experimental

and modelling works (i.e, Fasham et al. (1990)) considered a standard value 1 d$^{-1}$ for this parameter with ranges from 0.2 to 2.0 d$^{-1}$; we kept these values and even duplicate them to force and evaluate the system under different predation pressure conditions. We established the range of swimming speeds ($w_Z^{\max}$ values from 50 m to 320 m d$^{-1}$) to include the diversity of species that the mesozooplankton group could contain (Saiz et al., 2003; Kiørboe, 1993). It should be noted that this speed is the maximum that an organism can reach due to its physiological abilities, not meaning that it swims at that value constantly.

Because DVM depth preference is directly related to predator avoidance behavior, which is in turn closely related to body size





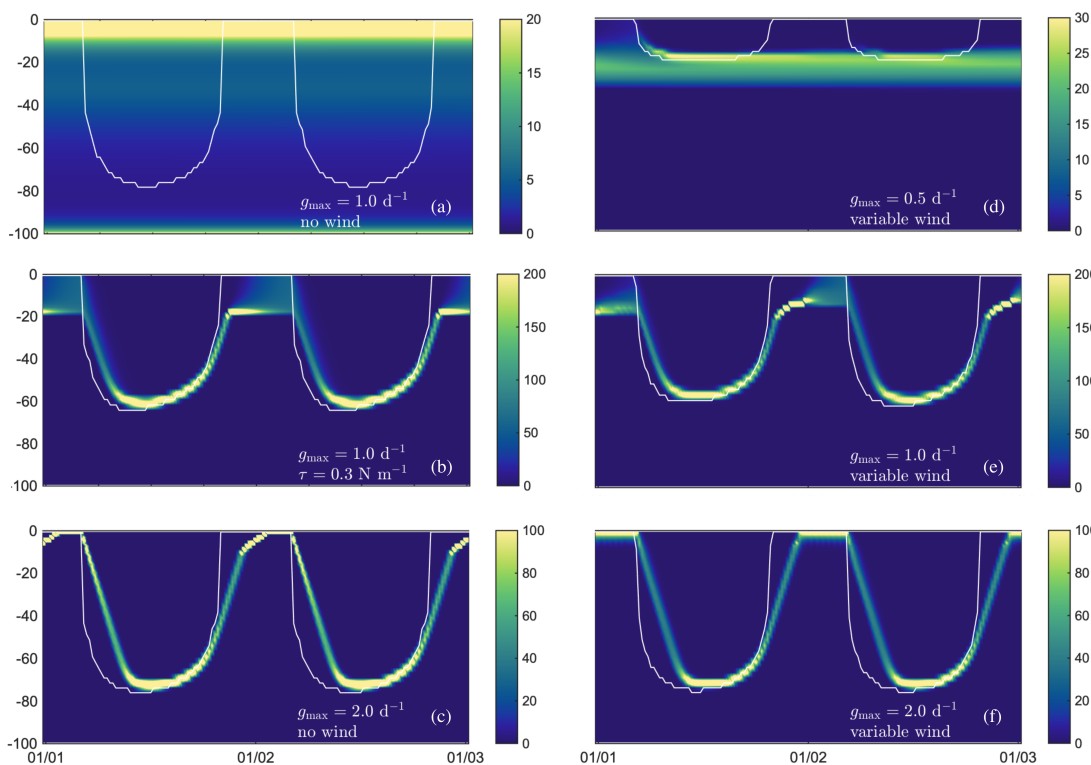

**Figure 4.** Representation of zooplankton concentration (in mgC m$^{-3}$) for two days (colored) and the isolume (white line) followed by the organisms when migrate in the water column for three different grazing rate (rows) values and turbulent regimes (columns). Parameter values are specified on the image and within the text (notice the different scales between panels).

and color (De Meester et al., 1999), we set three values for the critical irradiance $I_c$ = 0.01, 0.1 and 10 W m$^{-2}$ achieving different preferred depths following the optimal light intensity (Hays, 1995, 2003).

According to Fig. 6, for all cases explored, there is a more or less abrupt transition happening around a critical grazing rate that we call $g_{max}^{\star}$, from no migratory behavior ($\Omega = 0$) towards a stronger, more clearly defined one ($\Omega > 0$). In Fig. 6 *a* and *d*, in calm conditions (no wind), this transition occurs at $\sim 1.0\,\mathrm{d}^{-1}$ while it happens at $\sim 0.5\,d^{-1}$ when there is wind-induced mixing (Fig. 6 *b*, *c*, *e* and *f*). Beyond $g_{max}^{\star}$, $\Omega$ increases monotonically as a function of the swimming speed until it reaches a plateau, which corresponds to the situation where zooplankton migrates very efficiently in a way that is not perturbed by diffusion or grazing pauses. The latter condition arises from the fact that for large grazing rate values, zooplankton consumes phytoplankton intensely so that it never accumulates beyond $P_{min}$. Naturally, as turbulent diffusion increases, zooplankton patches are diluted, opposing to vertical migration is counteracted, which decreases the maximum value $\Omega$ reaches. The small values reached by





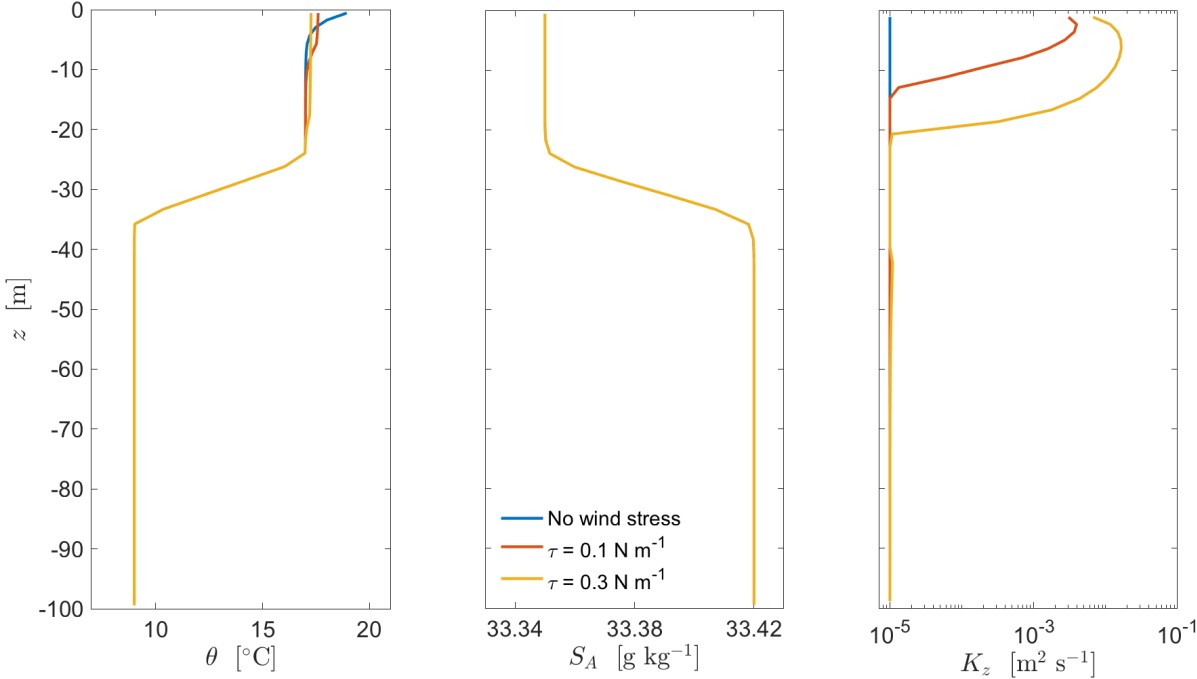

**Figure 5.** Vertical profiles for temperature, salinity and turbulent diffusivity without wind stress (blue line) and with two different constant wind stresses (0.1 and 0.3 N m$^{-1}$, red and orange lines respectively).

this indicator, even in optimal conditions, result from the fact that values are averaged over a full seasonal cycle. When grazing rates are below $g_{\max}^{\star}$, the vertical migration is very weak, which means either that there is no vertical maximum (no patch) or that patches do not move vertically, irrespective of the swimming speed. The same pattern is observed for the other critical irradiance values tested (not shown). The indicator shows weaker values when turbulence conditions increase ($\tau = 0.1$ N m$^{-1}$

not shown) and the relationship between the mentioned parameters is not so evident (four lower panels).

     Coupling between phytoplankton and zooplankton is higher for low grazing rates, but is independent of the swimming speed in all turbulence and irradiance scenarios, as shown by the grazing efficiency indicator ($\Psi$). There is a *strong* coupling when the $\Psi$ is lower than 0.5 (white dotted lines in Fig. 7). We recall that this means that, on average, at least 50% of the zooplankton in the water column is co-located with phytoplankton concentrations greater or equal to $P_{\min}$. The intensity of coupling decreases

with increasing grazing rates, regardless of zooplankton swimming speed. For $P_{\min} = 0.35$ mmolN m$^{-3}$ (Fig. 7 panels *a, b* and *d*), the strongest coupling is obtained for values smaller than 1.5 d$^{-1}$ in calm conditions ($\tau = 0$) and $I_c = 0.10$ W m$^{-2}$, and lower than 1.0 d$^{-1}$ in the variable wind scenario. Nevertheless, a small increase in the coupling for rates close to 1.0 d$^{-1}$ occurs for $\tau = 0$ and with a lower critical irradiance ($I_c = 0.01$ W m$^{-2}$). When $P_{\min} = 1.4$ mmolN m$^{-3}$ (Fig. 7 panels *c, e* and *f*), larger values of $\Psi$ are obtained for grazing rates around 0.5 d$^{-1}$, with a lower coupling in the variable wind scenario for the

same values. When $I_c = 10$ W m$^{-2}$ (not shown), results are similar to those described above for $I_c = 0.1$ W m$^{-2}$. In scenarios





with constant wind forcing ($\tau = 0.1$ not shown and $0.3$ N m$^{-1}$ panel $e$), the indicator patterns are similar to those presented for the variable wind scenario. In comparison with the control model, where DVM is absent, there is by definition a perfect match between the phytoplankton and zooplankton. However, the value of the indicator is very low due to the fact that zooplankton in the control run grows on detritus and bacteria near the ocean floor, which is decoupled from phytoplankton. In the upper layer,

the coupling with phytoplankton is also less intense because zooplankton is more dispersed.

Carbon export below from the surface layer, where primary and secondary production happens, to the deeper ocean is indicated by $\Phi$ as the proportion of the total carbon that is found below 80 m. This indicator is consistently larger when $g_{max} < 1.0$ d$^{-1}$ in all experiments (Fig. 8), irrespective of the values prescribed to all other parameters tested. The contrast between the export for grazing rates higher than $\sim 0.5 - 1.0$ d$^{-1}$ increases as mixing increases in the surface layer (Fig. 8$b, c,$

$e$ and $f$). In contrast, for the mentioned grazing rates, $\Phi$ shows higher exportation values when the wind stress is absent (Fig. 8 $a$ and $d$). However, $\Phi$ decreases as the grazing rate increases, and this is more evident for intermediate turbulent regimes, i.e. $\tau = 0.1$ N m$^{-1}$ (not shown) and $0.3$ N m$^{-1}$ (Fig. 8 $b$ and $e$), when comparing with variable wind forcing, for which there is a larger export for the same conditions (Fig. 8 $c$ and $f$). Some particular cases stand out in this pattern, when the swimming speeds are higher than 150 m d$^{-1}$ and the grazing rates are also large ($g_{max} > 2.5$ d$^{-1}$), the carbon export starts to be considerably

important again.

## 4    Discussion

Despite the complexity and remaining unknowns about zooplankton individual and collective swimming behavior, there is a large body of evidence that zooplankton aggregate at depth during the day and near the surface during the night, a phenomenon called diel vertical migration (Dodson, 1990; Doney and Steinberg, 2013; Lampert, 1989; Ringelberg, 2010). The parameteri-

zation we propose is meant to remain simple, to rely on mechanistic rules representing, i.e. that the behavior can be rooted to plausible causes such as visual predator avoidance (Hays, 2003; Ringelberg, 2010), in order to simulate the main characteristics of DVM in the ocean. The results are broadly in accordance with empirical and experimental works as well as with acoustic data where similar patterns were observed for zooplankton DVM (Greene et al., 1998; Skjoldal et al., 2013; Ringelberg, 2010).

The parameters that are directly involved in the present model, namely the maximum swimming speed, the critical light

level, and the minimum phytoplankton concentration, have been selected to broadly represent copepods' behavior in coastal environments. We wanted the model to be able to reproduce a DVM that obeys to a given light level in any conditions of irradiance, and we explored how this DVM was modified by environmental conditions, and affected the ecosystem response. Therefore, our model could be widely applied, for different light conditions and phytoplankton distribution in the water column, and even considering other organisms with different sizes and swimming speeds.

Sensitivity studies to parameters directly and indirectly involved in the migratory behavior show a wide range of ecosystem responses. The parameterization is able, in some conditions, to induce patchiness in zooplankton distribution, which is ubiquitous in marine ecosystems and in direct relation with food concentration (Folt and Burns, 1999). Patchiness, partly indicated by $\Omega$, is obtained when zooplankton is able to swim faster than it is diluted by turbulence. This condition is achieved when


**Figure 6.** Indicator of the zooplankton diel vertical migration $\Omega$ computed as in Eq. 3 as a function of the maximum grazing rate and the maximum swimming speed, for $I_c = 0.01$ W m$^{-2}$ ($a$, $b$ and $c$) and $I_c = 0.1$ W m$^{-2}$ ($d$, $e$ and $f$) and for three different wind forcings: no wind ($a$ and $d$), a constant wind stress $\tau = 0.3$ N m$^{-1}$ ($b$ and $e$) and variable wind stress ($c$ and $f$). $P_{\min} = 1.4$ mmolN m$^{-3}$ for all panels. Red stars indicate runs that were used as examples in Fig. 4





**Figure 7.** Grazing indicator ($\Psi$) showing spatial coupling between zooplankton and phytoplankton for: no wind stress (*a* to *d*) and variable wind stress (*c* and *f*) and two different critical irradiance values, $I_c = 0.10$ W m$^{-2}$ (*a* and *d*) and $I_C = 0.01$ W m$^{-2}$ (*b, c, e* and *f*). Notice that $P_{min} = 0.35$ mmolN m$^{-3}$ for *a, b* and *d* panels and equal to $P_{min} = 1.4$ mmolN m$^{-3}$ for *c, e* and *f* panels. Red stars show examples of Fig. 4.



**Figure 8.** Carbon export indicator ($\Phi$) representing the last 20 meters of the water column for two minimum phytoplankton concentrations 0.35 and 1.4 mmolN m$^{-3}$ (left and right columns, respectively) and different wind forcings: no wind stress (*a* and *d*), constant wind stress 0.3 N m$^{-1}$(*b* and *e*) and variable wind stress (*c* and *f*). The critical irradiance is the same and equal to 0.01 W m$^{-2}$ for all panels. Red stars refer to the examples in Fig. 4.





the swimming speed $w_Z$ is larger than the *diffusive speed* given by $K_z/H$, where $K_z$ is the turbulent diffusivity and $H$ is a vertical scale over which the diffusion is effective (e.g. the mixed layer in our case). Since the swimming speed is controlled by the local availability of food (phytoplankton), the grazing rate strongly impacts the occurrence of DVM and the associated zooplankton patchiness.

The grazing rate is the parameter that exerts the strongest control on DVM in our study and, consequently, on its impact on the ecosystem response. From a modeling point of view, this means that special care must be taken to grazing if one wishes to add a DVM parameterization into a biogeochemical model. In a model with DVM, since zooplankton is not systematically coupled with phytoplankton, the specified grazing rate must be tuned as if it was an instantaneous parameters instead of a daily averaged quantity. The critical phytoplankton concentration $P_{\min}$, the optimum irradiance level $I_c$ and turbulence play

secondary but still significant roles on DVM, patchiness, carbon export and primary production.

Although there is substantial evidence on DVM behavior, only a few studies and models include this phenomena in their analysis of ecosystem dynamics. The effect that such behaviour can have on coastal marine ecosystems under analysis here is still unclear and far from having a unique pattern. For example, in this study, carbon export to the bottom layer is large for low grazing rates, further increasing under high turbulence regimes (Fig. 8). For situations in which there is a mismatch between

phytoplankton and zooplankton at high grazing rates ($\Psi$ indicator), carbon export increased via sinking and turbulence when considering the bottom layer. At the same time, migration behaviour reduces the growth of its own zooplankton population (Haupt et al., 2009), due to lower grazing on phytoplankton pressure, decreasing carbon contribution to the system. Despite the fact that our results did not show an intense export of carbon to the bottom, it should be noted that the present numerical experiments are generated over a complete year. If we had considered only to the short period of phytoplankton post-bloom,

the exported carbon values would be higher and in accordance with literature (Boyd and Stevens, 2002).

The comparison between the models with and without DVM allows us to infer about the importance of adding DVM for the interpretation of marine ecosystem dynamics. First, a model without DVM exhibits zooplankton growth at depths where these organisms are not encountered usually, including near the ocean's floor. Adding DVM avoids having that problem. Second, with DVM, phytoplankton biomass attains higher concentrations with the incorporation of zooplankton migration behavior,

while zooplankton biomass itself is reduced due to the shorter time spent feeding on the surface, therefore having less available resources for growth. The direct and unavoidable trophic coupling between primary and secondary producers that characterizes most models is disrupted and grazing rates prescribed to migrating zooplankton must be modified and tuned with respect to relevant metrics.

Furthermore, in models without DVM, permanent phytoplankton-zooplankton match greatly influences pelago-benthic cou-

pling. Carbon export is an issue of great importance in determining the role of the ocean in global change related processes, and this amount can be severely overestimated in models without DVM. As a result of this new fractional interaction in space and time, carbon export to depth also varies. The fraction of phytoplankton that is not consumed by zooplankton becomes part of the detritus found in the water column, where it can also be grazed by zooplankton (though in a lower proportion/preference than phytoplankton) but eventually settles towards deeper layers. While it is known that DVM favors carbon sequestration

(Ringelberg, 2010; Turner, 2002, 2015), few studies include it and consider its effects on the potential carbon sequestration at


the sea bottom. The different cases presented here illustrate this fact, but further studies in different configurations are needed to better understand its importance and constrain its representation.

# 5    Conclusion

Zooplankton diel vertical migration, even if parameterized in a simplified way, adequately represents plankton dynamics in
marine coastal ecosystem. This kind of behavior should therefore be incorporated into models simulating the behavior and the role ocean food webs in the ocean carbon cycle. Including DVM modifies trophic interactions and allows to better estimate ecosystems productivity and carbon export. Our results suggest that the zooplankton grazing rate and swimming speed parameters are particularly important for an accurate representation of the carbon export in coastal shallow marine ecosystems, as we proposed here. However, different aspects would require further insight. Zooplankton life cycle (e.g. diapause and rates),
species-specific behavior, physiological costs of swimming activity, feeding strategies, the ultimate causes of DVM and DVM at each development stage, among others, represent limitations to our approach. Taking the above-mentioned precautions into consideration, it is possible and necessary to include the DVM behavior in NPZD models to better understand the ecosystem dynamics. The validity of some assumptions presented here awaits future confirmation from laboratory experiments and field measurements.

*Code availability.*    The code developed and used in this study is available from the open-access repository https://gitlasso.uqar.ca/dumoda01/
gotm_ismer.git.

*Author contributions.*    ACN conducted the biogeochemical and ecosystem model simulations, and performed most of the analysis. DD was responsible for the numerical code modifications and indicators construction. IRS provided input on the theoretical and model outputs interpretations. All authors contributed to the manuscript.

*Competing interests.*    The authors declare that they have no conflict of interest.

*Acknowledgements.*    This research was funded by the NSERC Discovery Grant No. 402257-2013 to D. Dumont and the FRQNT Québec-Océan Strategic cluster. Simulations were carried out on UQAR's supercomputing cluster Mingan supported by the Canadian Foundation for Innovation. It was started as part of AN Bec.Ar scholarship and is presently part of her work as CONICET Doctoral Fellow. All authors are thankful to Québec-Océan for their support.





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
