# Peer review of "A zooplankton diel vertical migration parameterization for coastal marine ecosystem modeling"

_Biogeosciences, 2020_

## Referee Comment (RC1) · Anonymous Referee #1 · 25 Mar 2020

General Evaluation

The authors present a 1-D NPZD model of coastal ecosystems with diel vertical migration (DVM) in zooplankton. DVM is parametrised by an optimal light level and allowed to occur only when food availability is below a certain threshold. A sensitivity analysis is presented showing the relation between DVM and several model parameters and boundary conditions. The authors conclude that DVM increases C export and that C export will be severely overestimated without DVM.

As may already be clear from the last sentence of the previous paragraph, my main impression from this work is that it has not been thought through very thoroughly. The

main model set-up and assumptions appear either rather half-baked or unfounded. Another problem I have with this study is that I expect from every model study at least some attempt to compare the model quantitatively with observations, and this has not been done here. Also, one of the central assumptions in this model is that DVM occurs only at low food concentrations, but this assumption is only mentioned but not really discussed in terms of how realistic it may be or whether it has been observed. In fact, no evidence is presented for it. While this assumption could make sense below the photic zone, where it could help determining the daytime depth of the zooplankton, I think it is introduced here too simplistically, in a way in which it could also keep the zooplankton very close to the surface. Because of these deficiencies, and I list some more below, I consider this work far below the current state of the art and do not recommend publication.

Specific points

Abstract, line 3: "... most zooplankton species." DVM occurs in many but probably not most species, e.g., it is not known for many microzooplankton species.

P. 2, line 5: "The efficiency 5 of the biological carbon pump is regulated by zooplankton and micronekton vertical migration ..." Surely, other processes and phenomena also affect the efficiency of the biological C pump. For example, sinking of phytoplankton and aggregates and their stoichiometry.

P. 7, line 30: "parameter space" It remains unclear how many parameters the model has, and why the ones examined here were selected.

P. 8, line 23: "rC:N = 7 mmolC mmolN−1" Further above the authors give two C:N ratios for phytoplankton and zooplankton, citing Redfiled (1958) (who did not distinguish between phytoplankton and zooplankton). So this is inconsistent with the model set-up and could lead to a mass-balance violation. It is not clear whether the model was checked for mass conservation.
P. 8, lines 26-27: "... a non-realistic benthic zooplankton community ..." A benthic community is not necessarily unrealistic, although it is of course not zooplankton, as long as it undergoes no (vertical) motions. This seems to be the case in the simulations without DVM.

P. 9, lines 25–26: "When turbulent diffusivity is low enough, zooplankton aggregate into narrow patches (Fig. 4c and f ), while if it is strong enough, it will be mixed homogeneously (Fig. 4b and e)." It is not clear that this is what is going on between Fig. 4b and e. Both panels look very similar, with zooplankton being concentrated near the optimal isolume.

P. 11, line 3–4: "a more or less abrupt transition happening around a critical grazing rate that we call gmax, from no migratory behavior ($\Omega = 0$) towards a stronger, more clearly defined one ($\Omega > 0$)." I think this sentence highlights a major problem of this ms. The model makes a very strong assumption, namely that DVM stops once a certain amount of food is available, and no evidence is presented for this assumption. Then major conclusions are drawn based on this assumption. This is an example for such conclusions. The effect of the maximal grazing rate is solely due to this assumption because it determines whether the zooplankton can graze down their food below the critical level allowing for DVM.

P. 13, lines 7–8: "This indicator [export] is consistently larger when gmax < 1.0 d−1 in all experiments" These are the same conditions where DVM is suppressed, so this is the foundation for the conclusion that C export can be severely overestimated without DVM. But apparently this does not fit with the authors' view of DVM as a process favouring export, so they provide both, resulting in a self-contradictory conclusions section.

---

## Referee Comment (RC2) · Frederic Maps (Referee) · 25 Mar 2020

General comments

The authors present the design and results of a theoretical numerical study of mesozooplankton diel vertical swimming behaviour (DVM) in coastal ecosystems. They develop a Eulerian framework to study more specifically the consequences of interactions between some external forcing (i.e. vertical turbulent mixing, phytoplankton concentration and light levels within the water column) and inherent properties of mesozooplankton populations (i.e. grazing rates and maximum swimming speed) on the vertical distribution of planktonic biomass and organic fluxes. They develop original, yet simple,

indices to evaluates the emerging properties of their simulations. While it is currently a simple 1D water column setup, the authors made several assumptions and choices in their study's design in order to assess quantitatively the role of DVM on the organic carbon budget simulated by coupled bio-physical NPZD-type models.

While I think the current manuscript is a valuable contribution to marine ecology, I think that it remains very theoretical in the absence of any data to validate the output and, further, that its applicability to other numerical studies is hampered by a critical assumption of the authors: the "coastal" environment represented in their study has to be deep enough to allow for diel vertical migrations to be, in part, cued by light levels. It would be very valuable to also take care of the majority of the cases occurring in coastal areas where troughs, channels and basins are actually few and separated by shallow areas where the proposed mechanisms may not play the same role at all. The article as it is now could also benefit from improvements in both the presentation of the ideas and the writing in general.

Detailed comments

1. P2 L4: "... their high biological productivity" The authors should provide the numbers about the relative proportion of the ocean surface they represent vs the relative proportion of the new production they contribute to; it is always useful for this kind of generic introduction.

2. P2 L5: I think reducing the efficiency of the biological carbon pump to zooplankton DVM is problematic... The authors should be more thorough in their description of the carbon pump. It is all the more important since their study is probably most relevant for computing the carbon budget of coastal ecosystems...

3. P2 L17: did the authors actually mean "organic matter" instead of "nutrients" here?

4. P2 L26-29: The authors literature review about vertical swimming behaviour of zooplankton is lacking. I think they should read carefully at least these two papers:

Pinti et al. 2019 http://doi.org/10.1098/rspb.2019.1645 and mostly Sainmont et al. 2013 http://doi.org/10.1007/s12080-012-0174-0.

5. P2 L32: The authors should provide some values as example to understand what kind of "intense vertical mixing" they think of, because counteracting zooplankton vertical swimming behaviour implies quite high mixing!

6. P2 L35: "stratification" appears here for the first time... I am not convinced yet that the authors managed to explain the role of stratification on zooplankton DVM.

7. P3 L2: I do not understand how "migratory behavior of zooplankton" is different from "diel vertical migration" ? How come "migratory behavior of zooplankton" would not impact "diel vertical migration" ?

8. P3 L3: what about primary production, then? One important assumption of this study is that it implements fully coupled NPZD-type model of pelagic production. Why are the author disregarding top-down control in their objectives definition, while they discuss it eventually?

9. P3 L17: I really do not think the "heart of this paper" is about "elucidating the causes of this migration"! The authors should refocus their message around the second part of their proposition ("... establishing significant correlations...").

10. P3 L19: the opening sentence of this paragraph is really awkward. It should be reformulated.

11. P3 L29: for equation (1), even if we accept the authors assumption that zooplankton swimming speed roughly follows an hyperbolic tangent attenuation profile, it is regrettable that they did not provide any data (most likely to come from detailed acoustic studies) to at least empirically calibrate their CORE swimming speed function!

12. P4 L3: what is the value of the critical Pmin parameter in this example simulation (Fig. 1)? Pmin = 0?

13. Fig. 1 (again): a simple side panel showing the shape of the swimming function at noon over the whole water column would be very useful.

14. Fig. 2: as it is now, this figure is not very useful. And its legend is confusing... it looks like it represents the swimming speed at the different depths represented by the dashed line in Fig. 1, not as "a function of time for the light..."

15. Fig 3. This figure is really confusing... The distinction between space and time is unclear. For example, is there a connection between the euphotic and aphotic areas during the day? How do you decide about the "intensity" of the relationships?

16. P7 L1: the grazing function being so important to the analysis it would be better to provide it: for example, does "sigmoidal form" mean a Holling type III ? What is kg ?

17. P7 L7: "...between 0.2 and 20 mm". It should be clearly stated that the authors aimed at achieving one common parameterization over two orders of magnitude in size.

18. P7 L11: since Fig. 5 is described before Fig. 4, both should be swapped.

19. P7 L11: "restored" when? At the end of a calendar year? Why?

20. P7 L11-12: The authors should explain in more details why having similar mixed layer depths is an important requirement of their modeling set up.

21. P7 L13: why is it different than the two-week relaxation time from above?

22. P7 L15: Regarding vertical eddy diffusivity specifically, surface wind stress is an important component (especially in 1D water column setups), but what about the turbulent kinetic energy created by horizontal shears? This is typically overlooked in 1D simulations, unless there is some form of minimum background level applied throughout the water column. Did the authors consider this? If so, how?

23. P7 L19: How did the author select a priori the parameters to be tested? How did they avoid the risk of overlooking something unexpected?

24. P7 L30: please provide the parameter space explicitly: name of parameters, range values.

25. P7 L31: this "indicator" approach is very interesting!

26. P8 L5: again, please provide the actual value required for Kz to counter the given Wz_max tested! I am positive some values will be ruled out as impossible...

27. P8 L9: Why did the author establish this threshold of vertically integrated zooplankton biomass. Integrated abundance has nothing to do with aggregative behaviour in their simulations!

28. P8 L23: about the RC:N = 7 ; did not the authors state in the Methods that there were 2 distinct C:N ratios, one for phytoplankton and one for the rest ?

29. P9 L5: Fig. 5 did not show the functions phi, psy and omega ?!

30. P9 L15: the authors choices for the values are arbitrary and should be better motivated.

31. P10: Table 2; I think the experiment numbers are not used within the text, which is a waste...

32. P10 L6: "... based on the literature". This is NOT enough. What processes did you want to explore with these specific values you did sensitivity analyses for?

33. P10 L11: really confusing sentence.

34. P11 Fig. 4: I DO NOT understand the organization of the panels. Please refer explicitly to the letters a) through f). As it is now, it does not look like the result of a factorial design, and I do not know what was the rationale for showing these particular results... Is there any migration at all in a), by the way?

35. P11 L1: about the light levels (Ic) : and what about the visual capability of the migrating zooplankton? Can they detect 0.01 W m-2 ? Alternatively, are there organisms

that are actually "camouflaged" at a light intensity of 10 W m-2 ?

36. P12 L1: the averaging over a full seasonal cycle is a choice. Why did the authors do it? Why did they not focus on the productive season?

37. P12 L5: "... and the relationship between the mentioned parameters is not so evident" maybe so, but this is not really acceptable here, since it is the authors duty to tease them appart.

38. P13 L22: BE CAREFUL! I don't think any of the references here deal with "experimental" work!

39. P13 L30: This part of the discussion should be tied much more directly to the Eulerian framework used in this modelling study. Actually, all the results discussed here have a meaning only in this peculiar context.

40. P17 L4: I guess the maximum swimming speed is important too?

41. P17 L7-9: I do not understand the argument about instantaneous grazing rate. I would like the author to develop and clarify their idea.

42. P17 L9: I understand, though, that this parameter is useless in a configuration where there is no feed-back of zooplankton on phytoplankton concentrations, i.e. an offline coupling which remains rather common in 3D coupled models of phytoplankton-zooplankton models. This situation can occur when simulation fields from distinct models or in situ observations are used, or in situ data.

43. P17 L18: please quantify how "intense" the carbon export is.

44. P17 L20: the authors can certainly provide the numbers from the literature they think their results agree with.

45. P17 L25: But the DAILY grazing rate should/could be modified accordingly and increased (under certain constraints) to allow for a migrating organism to graze enough in a shorter period at the surface! This is certainly the essence of the asynchronous

night-time behaviour observed in some zooplankton species, i.e. individuals go up to feed until they are satiated, then go/sink down, go back up again if necessary and in any case manage to get what they need during this time period (e.g. Sourisseau et al. 2008 http://doi.org/10.1139/f07-179 )

46. P17 L30: ". . . global change related processes" which ones?

47. P17 L32: "proportion/preference" please avoid this kind of shortcuts and explain what you mean when you collate two distinct notions like that.

48. P18 L6: Since there are no data provided, I think that there is nothing in this article that provide evidence that a model including DVM "better" or more "accurate" estimates coastal marine ecosystem productivity. The authors have just showed that the resulting dynamics is different with and without DVM.

Typos / minor modifications

1. P1 L18: replace "and/or" by "and".

2. P2 L8-9 and throughout: remove "relatively" and "potentially". Please abstain from using such modifiers (adverbs); it just dulls the authors' thesis.

3. P2 L26: replace "one copepod specie" by "one copepod species"

4. P3 L1: replace ". . . dynamics with DVM" by ". . . dynamics including DVM"

5. P3 L2: replace ". . . to characterize if in which" by ". . . to characterize in which"

6. P3 L2: ". . . zooplankton impacts"

7. P3 L21: replace "relatively easy interpreted" by "interpreted clearly".

8. P6 Fig. 3 caption: in general, prefer "relationship" over "relation".

9. P3 L21-22: replace "Zooplankton swimming behavior we impose here. . ." by "Simulated zooplankton swimming behavior. . ."

10. P3 L23: remove "mainly"

11. P3 L27: replace "irraidance" by "irradiance"

12. P8 L12: replace "prominence" by "concentration".

13. From here on, I provide an annotated pdf version of the paper to help with typos and writing issues.

Please also note the supplement to this comment:
https://www.biogeosciences-discuss.net/bg-2020-10/bg-2020-10-RC2-supplement.pdf

───────────────────────────────

**Supplement:**

[revised manuscript text omitted]

---

## Referee Comment (RC3) · Anonymous Referee #3 · 26 Mar 2020

General Comments

This study uses a 1-D NPZD model with a theoretical parameterization of the zooplankton diel vertical migration (DVM) to infer its impact on coastal ecosystem and carbon export. Simulations cover a wide range of parameters to analyse the sensitivity of the DVM and its impacts to model parameters (e.g. grazing rate, optimal irradiance) and boundary conditions (e.g. winds conditions). The authors conclusion stress the importance of the grazing rate and the swimming speed to accurately represent the carbon export in coastal shallow marine ecosystems

I found the objective of this study difficult to identify. What is the overall goal of the study

? While the experimental set up seems sound, results from previous model studies including a DVM parameterization are not discussed, which makes its impossible to identify new scientific inputs from the present study. Only one is really cited (the 1D model of Bianchi et al. 2013) and not thoroughly discussed. As an example, in the latter study, the optimal irradiance (Ic) chosen was 1.10-3 W.m-2, an Ic that is not even in the range of the tested parameters while the authors acknowledge its utmost importance in "accurately" reproducing DVM. What would be the added value to 3-D biogeochemical models (e.g. see Bianchi et al, 2013b; Aumont et al. 2018) ? Moreover, the choice of a coastal set up with shallow waters but with only surface turbulence considered would have required some justification, particularly if one of the main message of the study is referring to "benthic zooplankton". Coastal region where there is no tides or internal waves that will generate turbulence/mixing above the seafloor are so common ? What is the rationale to justifiy the relationship between phytoplankton availability and DVM ? Finally, the authors claim that "the zooplankton grazing rate and swimming speed param- eters are particularly important for an accurate representation of the carbon export in coastal shallow marine ecosystems", but no observations whatsoever is given to backup this assertion. As a conclusion, "as is", this study is not put in the context of either modeling studies or observational studies. The parameterization chosen and the experimental set up is not really discussed either.

Few specific comments : p3 line 27: irradiance p8 line 9: Why restrict the analysis over a zooplankton biomass threshold ? p9 line 18: On fig 4d the isolume is quite shallow (because of high phytoplankton concentration, I guess) therefore there is no need for zooplankton to go deep, isn't it ? Is this what you meant by this sentence :"The grazing rate is not sufficiently large to deplete phytoplankton, which remains abundant enough to provide zooplankton for a reason (with respect to the parameterization) to remain at this depth." ?

---

## Author Comment (AC1) · 19 May 2020

We address the points below, referee comments are in bold text.

**General Evaluation**

**The authors present a 1-D NPZD model of coastal ecosystems with diel vertical migration (DVM) in zooplankton. DVM is parameterised by an optimal light level and allowed to occur only when food availability is below a certain threshold. A sensitivity analysis is presented showing the relation between DVM and several**

**model parameters and boundary conditions. The authors conclude that DVM increases C export and that C export will be severely overestimated without DVM. As may already be clear from the last sentence of the previous paragraph, my main impression from this work is that it has not been thought through very thoroughly. The main model set-up and assumptions appear either rather half-baked or unfounded.**

It might be clear to the reviewer but for us, it is not clear how our conclusion led the reviewer to the impression we did not think about the question thoroughly. We are willing to demonstrate our assumptions are neither half-baked nor unfounded.

**Another problem I have with this study is that I expect from every model study at least some attempt to compare the model quantitatively with observations, and this has not been done here.**

It is true that the purpose of science is to explain and sometimes make predictions about aspects of the observable world. Developing models definitely serves that purpose. However, there are multiple types of useful modeling studies and comparing model results with observations is one among others. The scientific literature is full of examples where some aspects of ecosystem dynamics are discussed only in the *model world*, with only qualitative references to observations. The study of Huisman et al. (2006) published in *Nature* demonstrates that a simple 1D model of primary production in a turbulent stratified ocean can be chaotic, which is one fundamental source of irregularities observed in multiyear records of phytoplankton blooms and associated nutrient distributions. Abraham (1998) is another example of an idealized modeling study exploring how heterogeneity can emerge out of simple ecological happening in a turbulent ocean. These studies are very important for many reasons, especially when the object is of overwhelming complexity, so overwhelming that observing its dynamics can sometimes be impracticable.

The zooplankton component within numerical models is rarely compared against field

observations because, unlike other parameters such as temperature or Chlorophyll *a*. This might be so because zooplankton observations do not usually have the resolution of the modeled zooplankton variables (temporally or spatially), because they are in different formats or units (species abundance rather than mass expressed in nitrogen units), or because they are inaccessible (Everett et al. 2017). This is something that the scientific community needs to address: gathering modellers and field researchers to work together to better link models and observations. But this is not an essential exercise for the modelling itself.

**Also, one of the central assumptions in this model is that DVM occurs only at low food concentrations, but this assumption is only mentioned but not really discussed in terms of how realistic it may be or whether it has been observed. In fact, no evidence is presented for it. While this assumption could make sense below the photic zone, where it could help determining the daytime depth of the zooplankton, I think it is introduced here too simplistically, in a way in which it could also keep the zooplankton very close to the surface. Because of these deficiencies, and I list some more below, I consider this work far below the current state of the art and do not recommend publication.**

Here, probably our manuscript was not clear enough, which led the reviewer to think that our model indicates that zooplankton only moves at low food concentrations. The parameterization proposed in the present work refers to a $P_{\min}$ value, which does not mean a *low* concentration of phytoplankton, but to a minimum threshold concentration at which the zooplankton responds by migrating. Thus "low" food concentration should mean that there is too little food to support the current predator population, which isn't the case here. As we exposed in the text (Sensitivity analysis section), we examined three threshold values, for three different "food" concentrations, depicting the concentrations found at different stages of phytoplankton succession: before, during and after the bloom, 0.35, 0.7 and 1.4 mmolN m$^{-3}$, respectively (Schalles 2006).

Pearre (1970, 1973, 1979c) also suggested that hunger was the principal proximal

cause of zooplankton migrations to the surface, and that light served principally as a synchronizer between the different species. This statement is taken up in Pearre (2003) and long discussed in his work. This referred work also mentions that in conditions of low food abundance, animals could be found near the surface even in daylight, but eventually, because of the risk of predation they escape to deeper water. In addition, the author states that different species' synchronous migrations would be a special case of vertical migrations initiated by particular, mostly critical, food conditions.

In this sense, low food conditions together with low irradiance values should force migrators to spend more time in surface waters. This is a result shown by our model for winter, when applying the $P_{\min}$ concept explained above.

In relation with zooplankton found *near to the surface*, some species have been reported as mostly restricted to the upper layer (< 30 m; Escribano et al. 2012, 2009; Escribano and Hidalgo 2000; Escribano 1998). In addition, modelled copepod distributions in low food conditions showed that the only option to face this situation was to maximize growth (grazing where food is available), resulting in a shallow depth location (Fiksen and Giske 1995).

The complexity of marine biogeochemical or ecological models can be organized along two major axes, the biogeochemical or ecological complexity and the physical complexity (Gruber and Doney 2018). In this context, the plankton model implemented here presents a biogeochemical approach close to the highest complexity levels (as it is a NPZD including a bacterial loop). Although rather simple, the NPZD model considers several nonlinear interactions, zooplankton playing a pivotal role. As the model is quite cheap in terms of computation, this allowed us to perform sensitivity tests on the different threshold values or on the intensity of the zooplankton vertical velocity (more than 1500 simulations were performed). Regarding the physical complexity of the ocean, GOTM allows for a representation of a one dimensional ocean that allows for the understanding of a coupled food web model. Both together provide a good representation for upper ocean ecology focused on lower trophic levels. In summary, we consider our

approach as a good compromise for tackling a ecological question of plankton ecology in a well constrained physical (oceanographic) framework.

**Specific points**

**Abstract, line 3: "... most zooplankton species." DVM occurs in many but probably not most species, e.g., it is not known for many microzooplankton species.**

We have changed the word "*most* zooplankton species" by "*exhibited by a large number* of zooplankton species". Our assertion is true for mesozooplankton (particularly copepods) for which DVM is widespread (Ringelberg 2010), but we will change it to avoid confusion.

**P. 2, line 5: "The efficiency 5 of the biological carbon pump is regulated by zooplankton and micronekton vertical migration ..." Surely, other processes and phenomena also affect the efficiency of the biological C pump. For example, sinking of phytoplankton and aggregates and their stoichiometry.**

Referee #2 also suggested to add more details about the biological carbon pump (BCP), although the objective of this work is to emphasize the role that zooplankton vertical migration plays in this phenomenon and not the process itself. We have modified this paragraph and added more details about others components of the BCP that could affect the carbon export to depth in the water column before reaching the bottom and being sequestered in the sea floor.

The new paragraph states: "The efficiency of the biological carbon pump within coastal regions involves a series of carbon transformations (Fennel et al. 2018). These processes include the production of organic matter from inorganic carbon by phytoplankton, its consumption by primary consumers (zooplankton), the sedimentation of carcasses and dead organisms, as well as the formation of marine aggregates (Honjo et al. 2014). At the same time, it is argued by many studies that the active swimming of

zooplankton and micronekton regulates and even facilitates the carbon export (Bianchi et al. 2013, Steinberg et al. 2000, Steinberg et al. 2002, Tutasi and Escribano 2020). Zooplankton, apart from transferring energy to higher trophic levels, plays a central role in the active transport of dissolved and particulate organic matter to depth (Ducklow et al. 2001). The production of feces is particularly efficient for exporting carbon to the ocean's floor as fecal pellets sink fast, in the order of $1\text{-}10^2$ meters per day. The rate and timing at which they are produced is thus an important factor modulating the efficiency of the carbon sedimentation (Turner 2015)."

**P. 7, line 30: "parameter space" It remains unclear how many parameters the model has, and why the ones examined here were selected.**

In order to clarify this aspect, we decided to incorporate an Appendix section where all the model parameters are defined, which a recall from Burchard et al. (2006). The same list of parameters are used in the present study. The specific parameters presented in Table 1 are those that have been modified, added with our DVM parameterization, and varied in the sensitivity analysis.

**P. 8, line 23: "rC:N = 7 mmolC mmolN " Further above the authors give two C:N ratios for phytoplankton and zooplankton, citing Redfield (1958) (who did not distinguish between phytoplankton and zooplankton). So this is inconsistent with the model set up and could lead to a mass balance violation. It is not clear whether the model was checked for mass conservation.**

We thank the reviewer for pointing out that mistake. The text was changed to be in accordance with the description given in section 2.2 Biogeochemical model. We removed this citation and we included two others that are best suited for this, namely Fasham et al. (1990) and Anderson (1994). The model was checked for mass conservation, which will be stated explicitly in the text.

**P. 8, lines 26-27: "... a non-realistic benthic zooplankton community ..." A benthic community is not necessarily unrealistic, although it is of course not zooplank-**

**ton, as long as it undergoes no (vertical) motions. This seems to be the case in the simulations without DVM.**

The original version of the biogeochemical model we use here is based on the seminal work of Fasham et al. (1990), who aimed at representing the nitrogen cycle and plankton dynamics in a depth-integrated mixed layer sitting on top of a nitrogen reservoir. Kuhn and Radach (1997) coupled this model to a one-dimensional physical model, aimed at representing the pelagic planktonic ecosystem. This idea was later followed by Burchard et al. (2006) who applied the model in a realistic scenario. However, none of these works focused on zooplankton. One key aspect of the model is that zooplankton is able to graze on phytoplankton, detritus and bacteria. When detritus sink and accumulates on the seabed, zooplankton growth rate can become significant producing unrealistic zooplankton concentrations near the bottom. This is of course an artefact that has gone unnoticed in these studies, simply because it was not the focus of the paper. To avoid the growth of such zooplankton, other strategies can be employed, like removing detritus as soon as they reach the seabed, simulating carbon sequestration in sediments. When adding DVM, this "benthic zooplankton" growth can't happen anymore, which provides a mechanistic way to avoid this unrealistic phenomenon. An example of this was shown in Figure 4a.

To further clarify this, we propose replacing Figure 3 of our original manuscript by Figure 1 shown in this reply, where DVM is schematized in relation with the physical setting. The proposed caption would be : "Schematic illustration of the diel vertical migration in the context of a stratified marine environment. The left panel shows the relevant parts of the water column between the sea surface and the seabed. Wind forces turbulent mixing from the sea surface down to the pycnocline. The interior layer is characterized by low diffusivity ($K_z = 10^{-5}$ m s$^{-2}$) and the benthic layer is where detritus accumulates. Without DVM (central panel), zooplankton grows wherever there is food, which is predominantly phytoplankton in the euphotic layer, and detritus that accumulate near the seabed. With DVM (right panel), zooplankton swims toward a

preferred light level, sometimes fighting against turbulence, with occasional pauses wherever phytoplankton is sufficiently abundant. One consequence of DVM is that it never ventures below a certain depth and can't develop near the seabed."

**P. 9, lines 25–26: "When turbulent diffusivity is low enough, zooplankton aggregate into narrow patches (Fig. 4c and f ), while if it is strong enough, it will be mixed homogeneously (Fig. 4b and e)." It is not clear that this is what is going on between Fig. 4b and e. Both panels look very similar, with zooplankton being concentrated near the optimal isolume**.

We recognize that panels *b* and *e* look very similar. We wanted to explore the differences between constant wind scenarios with one where the wind forcing is realistically variable in time. What results show (Fig. 4 but also Figures 6 to 8) is that there aren't much differences between the constant high wind scenario and the chosen variable wind scenario. What's to remember though is that realistic values of turbulent diffusivities can be dilute zooplankton even though it swims up to 320m d$^{-1}$. In response to Referee #2, we added some quantitative analysis about what diffusivity can overcome what swimming speed.

We've decided to change panels in Figure 4 and present in the revised manuscript the numerical experiments listed in Table 2 to give more clarity to the Results Section (following Referee #2 comment).

**P. 11, line 3–4: "a more or less abrupt transition happening around a critical grazing rate that we call gmax, from no migratory behavior ($\Omega$ = 0) towards a stronger, more clearly defined one ($\Omega$ > 0)." I think this sentence highlights a major problem of this ms. The model makes a very strong assumption, namely that DVM stops once a certain amount of food is available, and no evidence is presented for this assumption. Then major conclusions are drawn based on this assumption. This is an example for such conclusions. The effect of the maximal grazing rate is solely due to this assumption because it determines whether the**

**zooplankton can graze down their food below the critical level allowing for DVM.**

We agree that the hypothesis that DVM is affected by food availability is a strong one and that we did not properly supported it. In his review, Pearre (2013) discusses the hunger/satiation hypothesis as a possible driver for vertical migration and cite multiple studies carried out over the past century that mention this effect, which is nonetheless very hard to confirm experimentally. The work by Bianchi et al. (2013), that we now describe in greater detail, uses that hypothesis in their DVM parameterization. We do not believe this assumption to be a major problem in our approach. We instead feel that including it as a possible driver and exploring its effect by way of a sensitivity analysis (the value of $P_{\min}$ is varied) can be valuable to determine how such a mechanism can impact a light-driven DVM parameterization. One result that comes out of this is that the resulting vertical excursions depend on how fast can zooplankton feed, decrease the local food stock,here represented by the maximum grazing rate $g_{\max}$, and proceed to migrate again.

**P. 13, lines 7–8:** *This indicator [export] is consistently larger when $g_{\max}$< 1.0 d$^{-1}$ in all experiments* **These are the same conditions where DVM is suppressed, so this is the foundation for the conclusion that C export can be severely overestimated without DVM. But apparently this does not fit with the authors' view of DVM as a process favouring export, so they provide both, resulting in a self-contradictory conclusions section.**

The sentence *This indicator [export] is consistently larger when $g_{\max}$< 1.0 d$^{-1}$ in all experiments* is in the Results section and describes what is happening in Figure 8 and indeed, as we mentioned in the text, reflect *all* experiments under this condition only. When $g_{\max}$ is increased beyond that value that we called $g_{\max}^{\star}$, we can see that C export increases for the model without DVM as well as when $w_{\max}$ takes large values. What we want to conclude from this is that the grazing rate is a key parameter influencing carbon export estimation.

On the other hand, the sentence *Carbon export can be severely overestimated without DVM* is mentioned in the Discussion section, but it wasn't a conclusion derived from our study. In this paragraph, we were pointing out the importance of including DVM into ecosystem dynamics to calculate carbon budget and we referred to this, but we didn't include the references which stand for it.

Here is the modified paragraph we propose: "Furthermore, in models without DVM, permanent phytoplankton-zooplankton match greatly influences pelago-benthic coupling. Carbon export is an issue of great importance in determining the role of the ocean in global change related processes,such as increasing anthropogenic $CO_2$ and its consequence on reducing sea pH, and this amount can significantly differ when DVM is considered in the system (Aita et al. 2003, Gorgues et al. 2019, Hansen and Visser 2016, Morales 1999, Tutasi and Escribano 2020). As a result of this new fractional interaction in space and time,as we show in the present study, carbon export to depth also varies. The fraction of phytoplankton that is not consumed by zooplankton becomes part of the detritus found in the water column, where it can also be grazed by zooplankton (though in a lower proportion/preference than phytoplankton) but eventually settles towards deeper layers. While it is known that DVM favors carbon sequestration (Ringelberg 2010; Turner 2002, 2015), few studies include it and consider its effects on the potential carbon sequestration at the sea bottom. The different cases presented here illustrate this fact, but further studies in different configurations are needed to better understand its importance and constrain its representation."

Please also note the supplement to this comment:
https://www.biogeosciences-discuss.net/bg-2020-10/bg-2020-10-AC1-supplement.pdf

sea surface

active mixing layer

- - - - - - - - - - - - - -
- - - - - - - - - - - - - -

euphotic depth

interior layer
low mixing

- - - - - - - - - - - - - -

benthic layer

seabed

without DVM

with DVM

**Zooplankton grows wherever food is, on phytoplankton in the surface layer and on detritus that accumulates near the seabed.**

migrating depth
set by $I_c$

**Zooplankton feeds on phytoplankton only a portion of the day and never reaches the seabed**

phytoplankton

zooplankton

detritus

**Fig. 1.** Schematic illustration of the diel vertical migration in the context of a stratified marine environment.

---

## Author Comment (AC2) · 19 May 2020

We sincerely thank Frédéric Maps for the time devoted to this review and for his constructive comments.

**General Evaluation**

**The authors present the design and results of a theoretical numerical study of mesozooplankton diel vertical swimming behaviour (DVM) in coastal ecosystems. They develop a Eulerian framework to study more specifically the con-**

**sequences of interactions between some external forcing (i.e. vertical turbulent mixing, phytoplankton concentration and light levels within the water column) and inherent properties of mesozooplankton populations (i.e. grazing rates and maximum swimming speed) on the vertical distribution of planktonic biomass and organic fluxes. They develop original, yet simple, indices to evaluates the emerging properties of their simulations. While it is currently a simple 1D water column setup, the authors made several assumptions and choices in their study's design in order to assess quantitatively the role of DVM on the organic carbon budget simulated by coupled bio-physical NPZD-type models.**

**While I think the current manuscript is a valuable contribution to marine ecology, I think that it remains very theoretical in the absence of any data to validate the output and, further, that its applicability to other numerical studies is hampered by a critical assumption of the authors: the "coastal" environment represented in their study has to be deep enough to allow for diel vertical migrations to be, in part, cued by light levels. It would be very valuable to also take care of the majority of the cases occurring in coastal areas where troughs, channels and basins are actually few and separated by shallow areas where the proposed mechanisms may not play the same role at all. The article as it is now could also benefit from improvements in both the presentation of the ideas and the writing in general.**

Our study is said to apply to coastal environments for two main reasons. The first one is that our parameterization is designed to explicitly simulate the vertical migration, with high spatial and temporal resolutions, or at least higher than any other previous attempts to model DVM. Such a configuration is compatible with existing high resolution coastal ocean models, but less with global models that are still operating on low resolution grids (e.g. Aumont et al. 2018 uses a 3h time step and a vertical grid size of ca. 25 m.). The second reason is linked to the configuration we decided to use, having a depth of 100 m and a set of parameters that broadly come from studies applied to

the coastal ocean. For example, the water transmittance is set according to type-III Jerlov waters, which is somewhere between the clearest offshore waters and the more opaque tidally-driven estuaries. Coastal here is used in a broad sense, to exemplify the relatively shallow, highly biologically productive marine shelf areas. However, our study was meant to focus on processes more than on geography, and the sensitivity analysis we carry can inform about the model response in situations we didn't expected originally. We agree that this study can be further extended to these other situations. To further clarify this, we plan to modify the way we conceptualize the model by changing Figure 3 of our original manuscript by Figure 1 below. The proposed caption would be : "Schematic illustration of the diel vertical migration in the context of a stratified marine environment. The left panel shows the relevant parts of the water column between the sea surface and the seabed. Wind forces turbulent mixing from the sea surface down to the pycnocline. The interior layer is characterized by low diffusivity ($K_z = 10^{-5}$ m s$^{-2}$) and the benthic layer is where detritus accumulates. Without DVM (central panel), zooplankton grows wherever there is food, which is predominantly phytoplankton in the euphotic layer, and detritus that accumulate near the seabed. With DVM (right panel), zooplankton swims toward a preferred light level, sometimes fighting against turbulence, with occasional pauses wherever phytoplankton is sufficiently abundant. One consequence of DVM is that it never ventures below a certain depth and can't develop near the seabed."

The referee is right in that the coastal environment we aimed at representing in this study requires a certain depth for the light attenuation to modulate the parameterized behavior. We tried different $I_c$ to highlight the point that finally, light is an important factor and therefore in the corrected version of the manuscript we will explicitly inform that the water column should be deep enough to allow for vertical migration. As said above, light and physics in general would correspond to a shelf system. So, the “coastal” characteristics would be better constrained. In addition, there is evidence that the depth reached by the zooplankton can correlated with water column depth (Hamame and Antenaza 2009). In any case, what we were trying to represent here are the main

characteristics for zooplankton DVM and the patterns associated with it. This behavior, once the "deep enough" question for the shelf system is considered, is the same irrespective of the actual depth: the zooplankton will be deeper in the water column during daytime and near the surface at night.

There are many other processes related to depth that may affect zooplankton dynamics, such as differential feeding according to the different types of prey in shallow versus deep environments (Hamame and Antenaza 2009), or zooplankton lipid content (Hays et al. 2001). However, these questions are completely beyond the scope of the present work. Nonetheless we thank Prof. Maps to allow us clarifying the depth range we are considering.

**Specific points**

**1. P2 L4: "...their high biological productivity" The authors should provide the numbers about the relative proportion of the ocean surface they represent vs the relative proportion of the new production they contribute to; it is always useful for this kind of generic introduction.**

This is a good idea. We included the percentage values taken for cited publications on our revision to be sent.

New text would say: "These areas cover 7-10% of the global ocean, but net primary production in these regions accounts for 10-30% of total production and on average carbon fixation is 2.5 times more compared to the deep and abyssal ocean (Fennel et al. 2018; Bauer et al. 2013). In terms of both inorganic and organic carbon export, coastal waters represent a significant contribution (up to 50% and 80%, respectively) to global carbon burial, and therefore the *continental shelf pump*" term was introduced (Tsunogai et al. 1999). Understanding the role of plankton food webs in the coastal ocean is thus particularly important."

**2. P2 L5: I think reducing the efficiency of the biological carbon pump to zooplankton DVM is problematic... The authors should be more thorough in their description of the carbon pump. It is all the more important since their study is probably most relevant for computing the carbon budget of coastal ecosystems...**

We agree with the referee suggestion, as the biological carbon pump is central to this work. However, we never mentioned that the efficiency is reduced due to zooplankton DVM. We added more literature background to this paragraph in order to complete the biological carbon pump description (see answer for Referee #1 on comment: P2, line 5).

**3. P2 L17: did the authors actually mean "organic matter" instead of "nutrients" here?**

Yes. Sorry about this mistake and thank you for noticing it.

**4. P2 L26-29: The authors literature review about vertical swimming behaviour of zooplankton is lacking. I think they should read carefully at least these two papers: Pinti et al. 2019 http://doi.org/10.1098/rspb.2019.1645 and mostly Sainmont et al. 2013 http://doi.org/10.1007/s12080-012-0174-0.**

We appreciate the mention of these references, which are very interesting; one is quite recent (we added this literature on DVM parameterization section, where we think it's relevant). However, what we were trying to emphasize in this introductory paragraph was that, even if there are many theories and models aimed at representing vertical migration (e.g. Doney and Steinberg 2013, Hylander and Hansson 2010, Haupt et al. 2009, Sourisseau et al. 2008, Rhode et al. 2001, Lampitt et al. 1993, among others), only a *few* of them deal with carbon export (e.g. Aita et al. 2003, Aumont et al. 2018) and biogeochemical models commonly do not include zooplankton DVM, irrespective of the number of dimensions or the model complexity (e.g. Tanioka and Matsumoto 2018 Curchitser et al. 2013, Duktkiewicz et al. 2009, Denman and Peña 1999, Doney

et al. 1994, just to mention a few). Probably our intention was not clear, so we will add this in the introduction section.

**5. P2 L32: The authors should provide some values as example to understand what kind of "intense vertical mixing" they think of, because counteracting zooplankton vertical swimming behaviour implies quite high mixing!**

Values will be listed in the revised manuscript. For example, some results obtained by Genin et al. (2005) indicated that under strong vertical velocities (4-6 cm $s^{-1}$) small zooplankton has to swim at velocities of > 10 body lengths in order to counteract its effects and remain at the desired depth or just out of turbulence range. Gallager et al. (2004) found that the distribution of *Calanus* spp. was aggregated wherever the turbulent RMS velocity was below 0.5 mm $s^{-1}$ (approx. 43 m $d^{-1}$)or otherwise diluted.

Actually, the third paragraph of the discussion in the original manuscript discussed this aspect, and Figure 4 shows examples where zooplankton is diluted by turbulence despite active swimming. However, we will specify values of turbulent diffusivities in the revised manuscript where relevant.

**6. P2 L35: "stratification" appears here for the first time... I am not convinced yet that the authors managed to explain the role of stratification on zooplankton DVM.**

Shelves are characterised by spatially and temporally varying stratification that is highly relevant for their physical dynamics and the seasonal changes of their ecosystems. Stratification strength has a direct impact on vertical turbulent transport (e.g., of nutrients), thus influencing food availability, by affecting phytoplankton growth and dilution, which in turn modifies zooplankton behavior.

The meaning of stratification in the context of our study will further be clarified with a new Figure 3 (see Figure 1 in this reply).

**7. P3 L2: I do not understand how "migratory behavior of zooplankton" is differ-**

**ent from “diel vertical migration” ? How come “migratory behavior of zooplankton” would not impact “diel vertical migration”?**

Indeed, DVM and migratory behavior are the same in the paper context. We have rephrased the sentence regarding the paper goal and characterizations that we are looking for, as also suggested by the Referee #3 due to lack of clarity.

New sentence for study aim: "It is our goal in this study to present a mechanistic parameterization for zooplankton diel vertical migration that is driven by light and food availability and that is compatible with coupled physical-biogeochemical models, and to describe its impact on the model’s response for a broad range of parameter values."

**8. P3 L3: what about primary production, then? One important assumption of this study is that it implements fully coupled NPZD-type model of pelagic production. Why are the author disregarding top-down control in their objectives definition, while they discuss it eventually?**

Primary production is indeed a very important driver of the ecosystem response. We chose not to characterize it in detail for the seek of conciseness. Even though we deal with a 1D 7-compartment BGC model, the parameter space is multidimensional and it becomes quite challenging to characterize one model run with one number in order to compare it with other model runs. That is why we designed indicators that tell something about the ecosystem dynamics, at the expense of some details. The grazing indicator $\Psi$ is meant to describe how zooplankton is *trophically* and spatially coupled to phytoplankton, its main source of food.

**9. P3 L17: I really do not think the “heart of this paper” is about “elucidating the causes of this migration”! The authors should refocus their message around the second part of their proposition (“... establishing significant correlations...”).**

We have added some suggested references and rephrased the paragraph.

New paragraph: "Vertical migration has been studied considering zooplankton ontogeny or physiological factors (i.e. Aita et al. 2003, Batchelder et al. 2002, Maps et al. 2011), as well as following sensory and feeding modes under the game theory model approach (i.e. Sainmont et al. 2013, Pinti et al. 2019). These papers focused on elucidating the causes of DVM and some prey-predator interactions as a consequence of the migrating behavior or establishing correlations with environmental or physiological conditions."

**10. P3 L19: the opening sentence of this paragraph is really awkward. It should be reformulated.**

We have removed the highlighted part of this sentence which does not provide much information and generated confusion.

**11. P3 L29: for equation (1), even if we accept the authors assumption that zooplankton swimming speed roughly follows an hyperbolic tangent attenuation profile, it is regrettable that they did not provide any data (most likely to come from detailed acoustic studies) to at least empirically calibrate their CORE swimming speed function!**

Please refer to the general comments where we present some of the echograms that motivated this modeling study.

**12. P4 L3: what is the value of the critical Pmin parameter in this example simulation (Fig. 1)? Pmin = 0?**

This is not an example simulation. This figure illustrates the evolution of the swimming speed vector as a function of time $w_Z(z,t)$ for a given irradiance field (Eq. 1). The value of $P_{\text{min}}$ is irrelevant here since there is no phytoplankton. It would however correspond to a situation where $P_{\text{min}}$ is larger than any phytoplankton value, such that zooplankton always swims towards the optimal light level.

**13. Fig. 1 (again): a simple side panel showing the shape of the swimming function at noon over the whole water column would be very useful.**

Noted. We will add such a panel.

**14. Fig. 2: as it is now, this figure is not very useful. And its legend is confusing...it looks like it represents the swimming speed at the different depths represented by the dashed line in Fig. 1, not as "a function of time for the light..."**

This figure shows the *speed* at which the isolume rises in the water column as time evolve during one day. It was meant to show that even if zooplankton can swim at a speed that is greater than 200 m $d^{-1}$, say 1000 m $d^{-1}$, it won't have to swim at that speed since the isolume travels downward/upward at speeds much lower during most of the day. Swimming faster than 200 m $d^{-1}$, in this example, will only affect the behavior during a very small fraction of the day. Such analysis we think help constraining the maximum speed to avoid violating the CFL stability condition in Eulerian numerical models. We note here that Bianchi et al. (2013) used a speed of 3 cm $s^{-1}$ = 2592 m $d^{-1}$. We will either remove this figure or discuss this issue more clearly.

**15. Fig 3. This figure is really confusing... The distinction between space and time is unclear. For example, is there a connection between the euphotic and aphotic areas during the day? How do you decide about the "intensity" of the relationships?**

We will replace this Figure by Figure 1 (this reply) in order to clarify many aspects of the paper and respond to a number of very valid questions and comments from the referees already mentioned above. This new figure does not contain the 7 biogeochemical compartments. Instead, we will add Figure 2 belonging to Burchard et al. (2006) in the new Appendix section together with key model equations and model parameter values. The proposed caption for Fig. 2 is: "Schematic diagram of the biogeochemical model that is based on classic NPZD interactions (green) to which is a microbial loop (orange) for remineralisation. Variables are nitrate $N$, phytoplankton $P$, zooplankton $Z$, detritus $D$, labile dissolved organic nitrogen (LDON) $L$, bacteria $B$ and ammonium $A$. Note that zooplankton feeds on phyplankton, bacteria and detritus with preferences $\rho_1$, $\rho_2$ and

$\rho_3$, respectively."

**16. P7 L1: the grazing function being so important to the analysis it would be better to provide it: for example, does "sigmoidal form" mean a Holling type III ? What is kg?**

This function was described in Fasham et al. (1990) and we did not make any modifications to it. It could be attributed to a Holling type III, as there is no saturation term by the functional response to the prey abundance by the predator or functional limitation to process the food. However, in this model this is the result of the de-coupling during daytime between prey and predator. "Kg" is the half saturation constant (now represented by $k_3$ in our Appendix A).

**17. P7 L7: "...between 0.2 and 20 mm". It should be clearly stated that the authors aimed at achieving one common parameterization over two orders of magnitude in size.**

As we mentioned in the text, we aim at representing copepods group, and therefore mesozooplankton, which size range is 0.2-20 mm. As the reviewer says, this indeed spans over two orders of magnitude, and this was considered when we set the swimming speed range too, as dependent on organisms' sizes.

**18. P7 L11: since Fig. 5 is described before Fig. 4, both should be swapped.**

We will change the order and the respective numbers as the reviewer suggested.

**19. P7 L11: "restored" when? At the end of a calendar year? Why?**

Restoring means that at each time step, a quantity $\delta$ is added or removed to a tracer such that it gets closer to a prescribed value. The increment delta is proportional to the difference between the current value $c$ and the prescribed one $c_s$, multiplied by the ratio of the time step $\Delta t$ and a restoring time scale $\tau$, also called the relaxation time.

$$\delta c(z,t) = \frac{\Delta t}{\tau} \left[ c_s - c(z,t) \right] \tag{1}$$

This procedure is quite often used in models in order to avoid state variables to diverge because key processes are not represented.

**20. P7 L11-12: The authors should explain in more details why having similar mixed layer depths is an important requirement of their modeling set up.**

We will do so in the manuscript. We will remind that only the *target* mixed layer depth is fixed and that it may vary around that target depending on the level of turbulent diffusivity that is used (Table 2 specifies these values). Indeed, we decided to fix the *target* mixed layer depth to limit the number of degrees of freedom of our sensitivity analysis, and because we wanted to minimize (but not eliminate) the variability of the primary production from one run to another. Finally, by keeping this depth rather unchanged from one experiment to another, we are able to characterize the effect of the maximum vertical excursion of the zooplankton (determined by $I_c$) *relative* to the mixed layer depth.

**21. P7 L13: why is it different than the two-week relaxation time from above?**

We admit there is confusion here. Temperature and salinity profiles are restored (relaxed) to those of Figure 5 (salinity profile not shown but specified in the text) within two weeks (14 days = 1 209 600 s). Nitrate concentrations, on the other hand, are restored at a different time scale, in order to make sure that the deep reservoir of nitrate does not deplete and that the nitracline is preserved. This will be more adequately described in the revised version.

**22. P7 L15: Regarding vertical eddy diffusivity specifically, surface wind stress is an important component (especially in 1D water column setups), but what about the turbulent kinetic energy created by horizontal shears? This is typically overlooked in 1D simulations, unless there is some form of minimum background level applied throughout the water column. Did the authors consider this? If so, how?**

The turbulent diffusivity profile we used in all simulations is shown in Figure 5. This profile has two main layers: a surface layer influenced by wind-induced turbulence, and an interior layer with a background diffusivity that is set to $1 \times 10^{-5}$ m s$^{-2}$. This value is in agreement with the mean diffusivity observed in a tidally-driven estuary where shear and internal waves happen. See for e.g. Cyr et al. (2011, doi:10.1029/2011JC007359), and more specifically their Fig. 9c. In comparison, the value that is used for representing the deep ocean interior is typically one order of magnitude smaller, i.e. $1 \times 10^{-6}$ m s$^{-2}$. We do not explicitly resolve shear currents or internal waves, but our simulations include their average effects on tracers.

**23. P7 L19: How did the author select a priori the parameters to be tested? How did they avoid the risk of overlooking something unexpected?**

The parameters we use for the sensitivity analysis are those in direct relation with the DVM parameterization that we introduced and that are absent from a model without the parameterization, namely $P_{\text{min}}$, $I_c$ and $w_Z^{\text{max}}$. To this list we also added the mixed layer turbulent diffusivity on the basis that the DVM is represented as an advective process applied to the zooplankton, an Eulerian variable, that is also affected by vertical diffusion. Finally, we explored the effect of the maximum grazing rate $g_{\text{max}}$, which controls the rate at which zooplankton deplete phytoplankton whenever it finds it on its path and the rate at which sinking particulate matter (detritus) is produced. Exploring how the ecosystem response is impacted by these five parameters poses a challenge and our study is surely not exhaustive. However, we believe it can provide valuable insight to the community as to how should mechanistic parameterizations of DVM be represented and used in Eulerian biogeochemical models.

**24. P7 L30: please provide the parameter space explicitly: name of parameters, range values.**

This part is described in Table 1, where the parameters in relation with DVM parameterization are listed as well as the range of values used in the sensitivity analysis.

We also added a complete list of all parameters and their values involved in the model (Burchard et al. 2006) in a new Appendix section.

**25. P7 L31: this “indicator” approach is very interesting!**

Thanks. After obtaining multiple model outputs (more than 1500), we decided that this was a necessary strategy.

**26. P8 L5: again, please provide the actual value required for Kz to counter the given Wzmax tested! I am positive some values will be ruled out as impossible...**

We discussed this in P17 L1-4 but didn’t provide any numbers. Here is what we propose to add: "Assuming a patch of zooplankton gathered around the optimal isolume due to swimming has a decay scale of the order of $\Delta z = 2$ m, the diffusivity value that is necessary to counteract the convergence due to swimming must be greater then $w_Z \Delta z$. Hence, zooplankton needs to be able to swim at a speed of 1 cm s$^{-1}$=864 m d$^{-1}$ for it to remain grouped in a patch when the diffusivity is $2 \times 10^{-2}$ m$^2$ s$^{-1}$, which is the maximum value reached in the mixed layer in our simulations. On the other hand, in the less turbulent waters ($2 \times 10^{-5}$ m$^2$ s$^{-1}$), a speed of only $5 \times 10^{-3}$ cm s$^{-1}$=4 m d$^{-1}$."

It is thus very possible for turbulence, in the mixing layer, to counter the maximum swimming speed used in this study, i.e. 320 m d$^{-1}$. This value corresponds roughly to the maximum physiological speed that an individual organism (belonging to the mesozooplankton) could reach. However, it does not mean that this constant speed is maintained. Therefore, when we talk about the possibility of being *mixed* or *diluted* within the mixing layer, we must take into account the net speed ($w_Z(z,t)$) that the zooplankton uses in both (upward and downward) movement. For example, Genin et al. (2005) found relationships between zooplankton vertical swimming speed and current velocities in up-welling zones, where in a current of 0.56 cm s$^{-1}$ zooplankton displacement is around 0.11 cm s$^{-1}$ (approx. 95 m d$^{-1}$).

**27. P8 L9: Why did the author establish this threshold of vertically integrated**

**zooplankton biomass. Integrated abundance has nothing to do with aggregative behaviour in their simulations!**

The indicator is based on the function *findpeaks*, which is a measurement of how a value in a given cell differs from the background value, irrespective of its absolute value. A threshold on the vertically integrated value was set to avoid identifying peaks when zooplankton concentrations are too low. This value was chosen after some tests were done to represent pre- to post bloom conditions.

**28. P8 L23: about the RC:N = 7 ; did not the authors state in the Methods that there were 2 distinct C:N ratios, one for phytoplankton and one for the rest?**

Yes. This was a contradictory mistake and was corrected. As the referee mentioned, this was described in section 2.2 Biogeochemical model and effectively there are two different C:N ratios, one for zooplankton (7:1) and other for the remaining state variables (6.6:1).

**29. P9 L5: Fig. 5 did not show the functions phi, psy and omega ?!**

No, we will rephrase this part to clarify this problem in our revised version. Figure 5 represents the vertical profiles for potential temperature, salinity and turbulent diffusivity in relation with the different turbulent regimes tested to forcing the model (and detailed in Table 2).

**30. P9 L15: the authors choices for the values are arbitrary and should be better motivated.**

We decided to present here just some of all the outputs we have obtained for this section, as they represent different patterns for zooplankton distribution within the water column. After this and other referees comments, we will change the panels for Figure 4.

**31. P10: Table 2; I think the experiment numbers are not used within the text, which is a waste...**

We have now added the experiment list in Table 2 and we will reorganize the text accordingly. We think that this new arrangement will give clarity to the text so that ideas are better presented in the section under the title "Examples of idealized experiments".

**32. P10 L6: "...based on the literature". This is NOT enough. What processes did you want to explore with these specific values you did sensitivity analyses for?**

This sentence opens the paragraph in which we explain how the choice of values in informed by what is found in the literature. Here is what we propose to better explain our choices: "The $P_{\text{min}}$ parameter is not something we can measure. It represents the degree to which zooplankton migration will be influenced by the presence of food. Values for $P_{\text{min}}$ were thus taken in the interval 0.35 to 1.4 mmolN m$^{-3}$, which is expected to influence the kick-off moment of this behavior in early-bloom conditions. Zooplankton grazing rate is a parameter that has been studied for decades. Were we chose different values for $g_max$ recognizing that in a model without DVM, zooplankton is spatially collocated with phytoplankton most of the time. In this situation, the instantaneous grazing rate also corresponds to the daily averaged grazing rate. When DVM is added, zooplankton is decoupled from the phytoplankton at least of portion of the day. The daily averaged rate can thus be very different from the instantaneous rate. Important experimental and modelling works (i.e, Fasham et al. 1990,Møller et al. 2012) consider a standard value 1 d$^{-1}$ with ranges from 0.2 to 2.0 d$^{-1}$. Here we used this range and extended it to 4.0 d$^{-1}$ in order to better characterize the ecosystem response in the limit of large values."

**33. P10 L11: really confusing sentence.**

See previous comment for a reformulation.

**34. P11 Fig. 4: I DO NOT understand the organization of the panels. Please refer explicitly to the letters a) through f). As it is now, it does not look like the result of a factorial design, and I do not know what was the rationale for showing these particular results... Is there any migration at all in a), by the way?**

We are sorry about the confusing organization. We will change Figure 4 panels to be in accordance with the numerical experiments proposed in Table 2. There is no migration in panel *a*, this is the control model (proposed by Burchard et al., 2006) where no DVM was present. We wanted to show through this panel the concentration near the bottom that characterizes the zooplankton (that we called "benthic zooplankton") and compare the water column distribution of this group after adding DVM to the model. In addition, we will also show different outputs for the numerical experiments (all listed in Table 2).

**35. P11 L1: about the light levels (Ic) : and what about the visual capability of the migrating zooplankton? Can they detect 0.01 W m-2 ? Alternatively, are there organisms that are actually "camouflaged" at a light intensity of 10 W m-2?**

Although these are interesting points, sensory and physiological zooplankton aspects are not contemplated within this model. This is one aspect that we highlight in the conclusion section as a limitation of the model and which is probably important to be modified or added in the future. Also note that light intensity within the water column also changes with phytoplankton shading and turbidity thus the optimal depth targeted by zooplankton will not be the same over a seasonal cycle.

**36. P12 L1: the averaging over a full seasonal cycle is a choice. Why did the authors do it? Why did they not focus on the productive season?**

The model is configured to represent coastal areas of temperate seas, that is, in these cases there are usually two blooms throughout an annual cycle. Therefore, choosing only the time of spring bloom would indeed be a choice, but it would leave out the second phytoplankton peak, which, although usually less important, can still contribute to carbon export.

**37. P12 L5: "...and the relationship between the mentioned parameters is not so evident" maybe so, but this is not really acceptable here, since it is the authors duty to tease them appart.**

The last sentence of this paragraph should read like this: "The migratory indicator $\Omega$ is weaker when turbulence is stronger in the mixing layer, and the dependence on $I_c$ become marginal".

**38. P13 L22: BE CAREFUL! I don't think any of the references here deal with "experimental" work!**

We have changed this sentence and edited the corresponding references: "The results are in accordance with empirical (i.e. Peterson et al. 1990, Ward et al. 1995), and experimental works (González et al. 1994), acoustic data (Ashjian et al. 1998, Falk et al. 2008, Mutlu et al. 2002, Record et al. 2006), as well as with modelling approaches where similar patterns were observed for zooplankton DVM in coastal waters (Greene et al. 1998, Skjoldan et al. 2013, Ringelber 2010)."

**39. P13 L30: This part of the discussion should be tied much more directly to the Eulerian framework used in this modelling study. Actually, all the results discussed here have a meaning only in this peculiar context.**

We agree with this comment. The discussion will remind this important element of context and conclusions will be steered towards Eulerian biogeochemical modeling issues.

**40. P17 L4: I guess the maximum swimming speed is important too?**

This is indeed important. We will add some quantitative values to illustrate this and deepen the discussion (see comment 26 above).

**41. P17 L7-9: I do not understand the argument about instantaneous grazing rate. I would like the author to develop and clarify their idea.**

This is explained in the response to comment 32 and it will be added in the discussion. In a model without DVM, the zooplankton is coupled to its food source most of the time. In this case, the daily averaged grazing rate is practically equal to the grazing rate that applies instantaneously at any moment during the day. On the other hand, where

zooplankton migrates, it grazes only a portion of the day, when it is spatially coupled to phytoplankton. The rate at which it does, generally during the night when it is near the surface, can then be significantly different than the daily averaged value. There is thus a distinction between the *instantaneous* grazing rate $g(t)$, the one that is given by the following equation and controlled by $g_{\mathsf{max}}$

$$g(t) = \frac{g_{\mathsf{max}} \rho_i c_i^2}{k_3 \sum_{j=1}^{3} \rho_j c_j + \sum_{j=1}^{3} \rho_j c_j^2} (Z + Z_{\mathsf{min}}) \quad i = 1, ..., 3. \tag{2}$$

and the *daily averaged* grazing rate $\bar{g}$ computed as

$$\bar{g} = \frac{1}{T} \int_0^T g(t) dt \tag{3}$$

where $T = 24$ h. Note that $\rho_i$ are preferences for the three types of food $c_1$ (phytoplankton), $c_2$ (detritus) and $c_3$ (bacteria). $Z$ is zooplankton and $Z_{\mathsf{min}}$ is a small value preventing the growth rate of being null when $Z = 0$. When $g(t)$ is constant over a 24-h cycle, then $\bar{g} = g_{\mathsf{max}}$. This aspect become relevant when one uses laboratory or experimental data since it becomes crucial to know if the measured rate corresponds to an instantaneous rate in the presence of food or an average rate over some integrated period of time. Eq. 2 will be presented in the new Appendix.

**42. P17 L9: I understand, though, that this parameter is useless in a configuration where there is no feed-back of zooplankton on phytoplankton concentrations, i.e. an offline coupling which remains rather common in 3D coupled models of phytoplankton zooplankton models. This situation can occur when simulation fields from distinct models or in situ observations are used, or in situ data.**

We agree that the grazing rate has a significance only when zooplankton and phytoplankton are coupled, like it is the case in a biogeochemical model. We will consider discussing this in the broader context of 3D offline models.

**43. P17 L18: please quantify how "intense" the carbon export is.**

We believe that quantifying the carbon export in the context of our study wouldn't adequate. We stress that it is not our intent to relate with observations or to provide conclusion as to whether our results improve our capacity to simulate *reality*. This would imply tuning our model to a particular situation, calibrate it and validate it. We will instead recall that we want to study the response of an ecosystem model when a mechanistic parameterization of DVM is introduced, and provide insight processes, interactions and the importance of some parameters.

**44. P17 L20: the authors can certainly provide the numbers from the literature they think their results agree with.**

See previous comment.

**45. P17 L25: But the DAILY grazing rate should/could be modified accordingly and increased (under certain constraints) to allow for a migrating organism to graze enough in a shorter period at the surface! This is certainly the essence of the asynchronous night-time behaviour observed in some zooplankton species, i.e. individuals go up to feed until they are satiated, then go/sink down, go back up again if necessary and in any case manage to get what they need during this time period (e.g. Sourisseau et al. 2008 http://doi.org/10.1139/f07-179 )**

See comments 32 and 41 about grazing rates.

**46. P17 L30: "...global change related processes" which ones?**

We now specify in the sentence just some of them which are relevant in the present study:"... such as increasing anthropogenic $CO_2$ and its consequence on reducing sea pH,..."

On the basis of the results presented in Laws et al. (2000), warming the surface waters of the ocean would be expected to decrease ef ratios (new production/total production = export production/total production) in the more productive parts of the ocean. Under

increased temperatures, decomposition rate of the organic matter (OM) in surface waters might be accelerated, and a higher stratification would further prevent OM to sink to depth before being complete degraded.

**47. P17 L32: “proportion/preference” please avoid this kind of shortcuts and explain what you mean when you collate two distinct notions like that.**

We’ve chosen "preference", as we were trying to emphasize that even if detritus is available in the water column, zooplankton have predilection for phytoplankton as primary food source.We will clarify the different passages in the text to indicate the preference for phytoplankton over detritus.

**48. P18 L6: Since there are no data provided, I think that there is nothing in this article that provide evidence that a model including DVM "better" or more “accurate” estimates coastal marine ecosystem productivity. The authors have just showed that the resulting dynamics is different with and without DVM.**

We agree with this comment. It reflects however that we failed to clearly state the objectives of the paper and put the results in the proper context. We will thus rework the discussion accordingly. Nonetheless, showing that results with and without DVM are different is important in the perspective that it is quite well known that DVM happens in some if not most systems. Integrating this process in biogeochemical models might be as important as incorporating the microbial loop in simulating remineralization and the associated regenerated primary production. It is key however to understand and document the implications of doing so instead or blindly introducing a process in a complex model and comparing the output with observations.

**Typos / minor modifications**

We appreciate the time and the thoroughness with which the reviewer has read the manuscript and highlight these typing and grammatical errors. Each of them was accepted and corrected for the revised version.

**1. P1 L18: replace "and/or" by "and".**

OK.

**2. P2 L8-9 and throughout: remove "relatively" and "potentially". Please abstain from using such modifiers (adverbs); it just dulls the authors' thesis.**

OK. We eliminate this kind of words, as the referee suggested.

**3. P2 L26: replace "one copepod specie" by "one copepod species"**

Done.

**4. P3 L1: replace "... dynamics with DVM" by "... dynamics including DVM"**

We have changed the word.

**5. P3 L2: replace "... to characterize if in which" by "... to characterize in which"**

The replace was made.

**6. P3 L2: "...zooplankton impacts"**

The "s" was added.

**7. P3 L21: replace "relatively easy interpreted" by "interpreted clearly".**

The replacement was made.

**8. P6 Fig. 3 caption: in general, prefer "relationship" over "relation".**

OK. The word has been changed in all the manuscript.

**9. P3 L21-22: replace "Zooplankton swimming behavior we impose here..." by "Simulated zooplankton swimming behavior..."**

The change was made. We recall that it is important to highlight the difference.

**10. P3 L23: remove "mainly"**

OK.Word removed.

**11. P3 L27: replace “irraidance” by “irradiance”**

OK.

**12. P8 L12: replace “prominence” by “concentration”.**

Word replaced.

**13. From here on, I provide an annotated pdf version of the paper to help with typos and writing issues.**

As we mentioned above, we have made all the necessary changes and corrected the typing problems.

**Have you any literature references about "Experimental" work on DVM? It looks awfully challenging...**

We know about some recent experimental work on *Artemia salina* (Houghton et al. 2018. Vertically migrating swimmers generate aggregation-scale eddies in a stratified column. *Nature* **556**. https://doi.org/10.1038/s41586-018-0044-z). However, the first studies for this phenomenon date back to the 80s, e.g. Bohrer (1980) who found through deep tank experiments, that the marine copepod *Calanus finmarchicus* spent more time at the surface when food concentration was low than when food was high. Cohen and Forward (2005c) also performed experimental work to study predator kairomone effects on marine zooplankton (*Calanopia americana*) behaviour. There is also literature for aquatic environments where this kind of experiments were performed in mesocosms (Leach et al. 2015).

Please also note the supplement to this comment:
https://www.biogeosciences-discuss.net/bg-2020-10/bg-2020-10-AC2-supplement.pdf

[Figure]

**Fig. 1.** Schematic illustration of the diel vertical migration in the context of a stratified marine environment.

[Figure]

**Fig. 2.** Schematic diagram of the biogeochemical model.

**Supplement:**

**General comments to all three referees**

We are thankful to the three reviewers for their thoughtful comments on the manuscript. After a careful reading of the reviews, it is clear that we will need to modify the text, some figures, as well as substantial parts of the manuscript itself to achieve more clarity. This will require some time, but in the meantime we wish to outline how we would like to respond, and to clarify some of the concerns raised by the referees. We already took care of small (i.e., typo) changes based on specific comments of the reviewers, and in the figure of the conceptual model.

Our modeling study was part of a multidisciplinary project (PROMESS project; see *Oceanography* **31**(4)) for which we gathered data in the San Jorge Gulf, located on the Patagonian Shelf, in Argentina. Many data are still unpublished as journal papers but available in two PhD and MSc theses. These data include echograms showing zooplankton vertical migration in the water column (see Figure 1). Moreover, a large number of observations on diel migratory patterns are also found in the published literature, such as for the continental shelf of Eastern United States (Ashjian et al. 1998), the inner Scotian shelf (Cochrane et al. 1991), the Gulf of Maine (Baumgartner and Fratantoni 2008), and the Black Sea (Mutlu 2002), just to name a few.

Our goal in this work is to dive into the representation of DVM in plankton models. We want to understand how introducing DVM as a behavioral trait of the zooplankton compartment of a widely-used NPZD model affects the response of this model, focusing especially on the carbon export in coastal waters. It is to this end that we establish a new, yet simple, parameterization for DVM in this less studied environments through models, keeping in mind that complexity often emerges from simple sets of rules. As Aumont et al. (2018) explicitly assert in their publication, there are so far two studies that represent DVM in biogeochemical models: Bianchi et al. (2013b) who present a 1D water column modeling study, and their own work, a 3D modeling study, both focusing on the open global ocean, characterised by deep environments. This last model was also implemented in Gorgues et al. (2019), where the migrating depth is such that it excludes the continental shelves and where they state that the bathymetry determines the diel movement of mesozooplankton. In contrast, our work is meant to be a tool for improving processes modelling within the pelagic ecosystems, which will hopefully serve the larger community interests of interpreting measurements and compile better observations on zooplankton group. In our humble contribution to the efforts of the larger community, our purpose is to look closer and better understand processes and their interactions.

As the majority of the scientific community agrees, zooplankton DVM plays a key role on carbon export, especially in coastal waters, such as shelves, estuaries and fjords, where large amounts of organic carbon are being buried every year. In addition, even if these environments (coastal shelves and seas) represent a small portion of the global ocean area, they account for 30 % of the total carbon globally buried. Given zooplankton's importance, and the widespread and ubiquitous nature of DVM, it is worth considering how to incorporate this process in models that are used to understand marine ecosystem dynamics.

Most plankton models are phytoplankton-centered. Arhonditsis and Brett (2004) found in published biogeochemical models that 95% of them validated their results against phytoplankton data, but less than 20% compared model output with zooplankton data. Also, in the relatively rare instances where zooplankton were considered in biogeochemical models, they were more poorly simulated than almost any other state variable (Arhonditsis and Brett 2004).

In this sense, our aim in revising the manuscript will be both to rewrite and restructure the text to clarify the general and specific objectives. We understand now that it is particularly important to expose this background context and to cover the above mentioned aspects when we describe our parameterization, our experimental set up, our results and the discussion. In summary, this study is intended to elucidate the impacts of introducing zooplankton DVM on plankton-based ecosystem dynamics and its carbon production and export in relation with physical driving forces such as turbulence and light. Additionally, significant efforts will be dedicated to improve the individual sections.

In addition to the general comments, perhaps most importantly, we will explain much more clearly the underlying background (regarding general comments of Referee #1 and #3), which is in relation with hunger-satiation hypothesis largely discussed by Pearre (2003) for zooplankton migrating groups in aquatic and marine ecosystems. This is a strong basis on which our DVM parameterization as a function of a critical phytoplankton concentration is laid out, which is strongly criticized by Referee #1.

[Figure]

**Figure 1.** Example of echograms obtained for the San Jorge Gulf (Patagonia, Argentina). Upper panel: during the day-night transition for 11st August 2011 (EK500 and EY500, 200 kHz), taken from Alvarez Colombo (2013). Lower panel: 48 hs echogram (EK60, 120 kHz), adapted from Nocera (2018) and courtesy of Valeria Retana (PROMESS project).

Finally, we will also add an Appendix to address comments regarding parameters and parameter space, that are well explained in Burchard et al. (2006) and that are reused in the present work.

We thus ask the Editor to agree on our manuscript to be reviewed after the modifications we proposed and explained above are done, hoping it will then be acceptable for publication in BG.

---

## Author Comment (AC3) · 19 May 2020

We address the points below, referee comments are in bold text.

**General Evaluation**

**This study uses a 1-D NPZD model with a theoretical parameterization of the zooplankton diel vertical migration (DVM) to infer its impact on coastal ecosystem and carbon export. Simulations cover a wide range of parameters to analyse the sensitivity of the DVM and its impacts to model parameters (e.g. grazing rate, op-**

**timal irradiance) and boundary conditions (e.g. winds conditions). The authors conclusion stress the importance of the grazing rate and the swimming speed to accurately represent the carbon export in coastal shallow marine ecosystems.**

**I found the objective of this study difficult to identify. What is the overall goal of the study ? While the experimental set up seems sound, results from previous model studies including a DVM parameterization are not discussed, which makes its impossible to identify new scientific inputs from the present study. Only one is really cited (the 1D model of Bianchi et al. 2013) and not thoroughly discussed. As an example, in the latter study, the optimal irradiance (Ic) chosen was 1.10-3 W.m-2, an Ic that is not even in the range of the tested parameters while the authors acknowledge its utmost importance in "accurately" reproducing DVM. What would be the added value to 3-D biogeochemical models (e.g. see Bianchi et al, 2013b; Aumont et al. 2018)?**

We thank the reviewer for recognizing the qualities of our experimental set-up. We agree that the review of other modeling studies is not as thorough as it should. We will thus add text in our manuscript to fill that gap. Note that at the time the bulk of this work was done, Bianchi et al. (2013) was the only existing study. Since then, others have been published.

- The model of Bianchi et al. (2013) is the only existing model that explicitly simulate the migration of an Eulerian zooplankton compartment. Their parameterization is based, like ours, on the hypothesis that light is the main driver of DVM. They chose their isolume $I_c = 10^{-3}$ W m$^{-2}$ in order to comply with observations of the migration depth, a value that is kept unchanged in all their simulations. Zooplankton migrates at a maximum speed $w_Z = 3$ cm s$^{-1} = 2592$ m d$^{-1}$, set constant everywhere except when approaching the isolume depth or the food maximum (the value of the maximum is not prescribed), in which case the speed decreases linearly to zero over a scale of 50 and 20 m, respectively. We recall

here that our parameterization uses a vertical swimming speed that continuously scales as a function of the difference between the local irradiance and the optimal isolume $I_c$, and that zooplankton stops swimming when it encounters enough food, i.e. when the phytoplankton concentration is larger than $P_{\min}$. Instead of assuming one value is representative of Nature, we carry a sensitivity analysis to explore how the ecosystem may respond if these parameters vary, which we think is a sound scientific approach to study highly complex systems with uncertain parameters. We stress that those two parameters are very seldom measured directly, but depend on indirect observations. Values we chose for the maximum swimming speed (i.e. animals are far from the preferred isolume) represent a maximum swimming speed of three body lengths per seconds for mesozooplankton ($\mathcal{O}10^2$ m d$^{-1}$). Parameters driving DVM in the study of Bianchi et al. (2013) have been chosen somewhat arbitrarily, as they claim themselves.

- Aumont et al. (2018) present a model, later used by Gorgues et al. (2019), that implicitly simulate DVM by assuming that zooplankton migrate instantaneously. This strategy is used because both the vertical resolution and the time step are quite large (the vertical grid spacing is 25 m near the surface and the time step is $\Delta t = 10\ 800$ s$= 3$ h) in order to limit the computational costs implied by the global domain and the biogeochemical model. Their choice of isolume is made according to a "quick" comparison with acoustic data and discrepancies are explained by an improper representation of light attenuation. Moreover, their implicit representation does not allow turbulence to affect the vertical movements of zooplankton, like it is the case in our explicit Eulerian framework.

The added value of our study is specifically to simplify the physical configuration in order to explore the range of parameters that may affect the plankton dynamics in the presence of DVM. There is no evidence in the relevant literature that a preferred representation of DVM exist with a defined set of parameter values.

**Moreover, the choice of a coastal set up with shallow waters but with only surface turbulence considered would have required some justification, particularly if one of the main message of the study is referring to "benthic zooplankton". Coastal region where there is no tides or internal waves that will generate turbulence/mixing above the seafloor are so common?**

Figure 1 (this reply) and the associated caption will clarify and justify the choice of our model configuration and to what type of coastal ocean does it apply or refer. See response on comment 22 of Referee #2. The proposed caption for Fig. 1 is: "Schematic illustration of the diel vertical migration in the context of a stratified marine environment. The left panel shows the relevant parts of the water column between the sea surface and the seabed. Wind forces turbulent mixing from the sea surface down to the pycnocline. The interior layer is characterized by low diffusivity ($K_z = 10^{-5}$ m s$^{-2}$) and the benthic layer is where detritus accumulates. Without DVM (central panel), zooplankton grows wherever there is food, which is predominantly phytoplankton in the euphotic layer, and detritus that accumulate near the seabed. With DVM (right panel), zooplankton swims toward a preferred light level, sometimes fighting against turbulence, with occasional pauses wherever phytoplankton is sufficiently abundant. One consequence of DVM is that it never ventures below a certain depth and can't develop near the seabed."

**What is the rationale to justifiy the relationship between phytoplankton availability and DVM?**

There is large set of data supporting the relationship between food (phytoplankton) and zooplankton migratory behavior (DVM) resulting in the hunger-satiation hypothesis (Pearre 2003). In the mentioned work, and supported by more literature (references therein), phytoplankton availability is the principal trigger to zooplankton "upward" movement, and light could be considered as a secondary factor. Accordingly, in our model zooplankton maximum depth is in close relationship with phytoplankton concentration, low phytoplankton availability driving upward migration to the surface where

there is supposedly greater prey concentration.

**Finally, the authors claim that "the zooplankton grazing rate and swimming speed parameters are particularly important for an accurate representation of the carbon export in coastal shallow marine ecosystems", but no observations whatsoever is given to backup this assertion. As a conclusion, "as is", this study is not put in the context of either modeling studies or observational studies. The parameterization chosen and the experimental set up is not really discussed either.**

We think there is a lack of clarity in the text in this section from our side. The conclusion section is focused on the results obtained under this specific work, that is, for a 1D NPZD model representing a stratified marine environment (where tidal mixing is low). DVM effects could be quite different from those found in deeper ocean areas, but the associated patterns can match (Hamame and Antenaza 2009). We will rephrase this part of the section by putting an emphasis on how DVM affects biogeochemical interactions from a modeling point of view instead of concluding broadly on carbon export, which can't be easily deduced. We will further discuss other modelling exercises both in terms of the parameterization as well as on the results obtained in the respective modelling scenarios.

**Specific points**

**p3 line 27: irradiance**

Thanks. The typing error was fixed.

**p8 line 9: Why restrict the analysis over a zooplankton biomass threshold?**

We think there was a misunderstanding as the restrictive rule that we implemented for simulated zooplankton vertical migration was a phytoplankton minimum concentration threshold. This is described in the Equation 1 (DVM parameterization). The assumption that underlies this rule is the hunger-satiation hypothesis in vertical migration presented by Pearre (2003).

**p9 line 18: On fig 4d the isolume is quite shallow (because of high phytoplankton concentration, I guess) therefore there is no need for zooplankton to go deep, isn't it ? Is this what you meant by this sentence :"The grazing rate is not sufficiently large to deplete phytoplankton, which remains abundant enough to provide zooplankton for a reason (with respect to the parameterization) to remain at this depth."?**

(We've changed Figure 4 order as suggested by Referee #2, now is mentioned as Figure 5).

Yes, due to the chosen zooplankton grazing rate value = 0.5 d$^{-1}$, phytoplankton concentration never attain values below the imposed minimum concentration $P_{min}$= 1.4 mmol N m$^{-3}$ that induces zooplankton to move. Thus zooplankton does not migrate following phytoplankton and their maximum concentration is confined during day and night between 15 and 25 m.

Please also note the supplement to this comment:
https://www.biogeosciences-discuss.net/bg-2020-10/bg-2020-10-AC3-supplement.pdf
* * *
[Figure]

**Fig. 1.** Schematic illustration of the diel vertical migration in the context of a stratified marine environment.